# Stochastic Block Model-Aware Topological Neural Networks for Graph Link Prediction

**Yuzhou Chen**  *yuzhou.chen@ucr.edu*
*Department of Statistics*
*University of California, Riverside*

**Xiao Guo**  *xiaoguo@nwu.edu.cn*
*School of Mathematics*
*Northwest University Xi'an*

**Shujie Ma**  *shujie.ma@ucr.edu*
*Department of Statistics*
*University of California, Riverside*

**Reviewed on OpenReview:** *https://openreview.net/forum?id=FBjVSPAsgs*

## Abstract

Link prediction is an important learning task for graph-structured data and is indispensable to understanding graphs' properties. Recent works focus on designing complicated graph neural networks (GNNs) architectures to explore and capture various pairwise interactions among graph nodes. Most GNNs are based on combining graph structural and node feature information by iterative message-passing schemes. However, despite GNNs revolutionizing the field of graph representation learning, some thorny questions are raised concerning whether GNNs can simultaneously learn topological information, node-wise community memberships, block-to-block connection probabilities, and provide statistically rigorous uncertainty estimates. In this paper, we tackle these challenges and propose a novel stochastic block model (SBM)-aware topological neural networks, called SBM-TNN, that uses SBMs to infer the latent community structure of nodes from graph structures and uses persistent homology to encode higher-order information. Furthermore, we theoretically study the entrywise bound and asymptotic normality of the estimated edge probability matrix to quantify the uncertainty in statistical inference of the edge probabilities. Our extensive experiments for link prediction on both graphs and knowledge graphs show that SBM-TNN achieves state-of-the-art performance over a set of popular baseline methods.

## 1 Introduction

Graph data are ubiquitous throughout the natural and social sciences, e.g., many real-world objects can be represented by graphs, such as molecules, ecosystems, transportation systems, energy systems, citation networks, and internet networks (Sen et al., 2008; Li et al., 2018; Xia et al., 2021; Chen et al., 2023a). Tremendous advances in graph analysis have been achieved in recent years, especially in the field of geometric deep learning (GDL) (Defferrard et al., 2016; Bronstein et al., 2017; Zhang et al., 2020). In particular, graph neural networks (GNNs) have emerged as effective architectures for various prediction problems, e.g., node classification (Kipf & Welling, 2017; Veličković et al., 2018a; Hamilton et al., 2017), community detection (Chen et al., 2018; Shchur & Günnemann, 2019), and graph classification (Xu et al., 2018; Ying et al., 2018). specifically, GNNs are neural network architectures designed to handle graph-structured data. The fundamental idea behind GNNs involves treating the underlying graph as a computation graph and leveraging neural network primitives to generate node embeddings. The key processes involve message passing, propagating, and aggregating node features and graph structural information throughout the graph.

In GNNs, the graph convolutional layer builds upon the observed graph adjacency matrix, also called the connection matrix. The adjacency matrix can be viewed as a noisy version of an edge probability matrix with additive noises. In practice, graphs often contain communities, and thus the probability of an edge between any two nodes can depend on their group assignment, which is popularly modeled through the stochastic block model (SBM) (Holland et al., 1983) or its variant the degree-corrected stochastic block model (DCSBM) (Karrer & Newman, 2011). When a graph indeed has communities, we propose to use the estimated edge probability matrix to replace the adjacency matrix for graph representation learning. We estimate the edge probability matrix using the spectral clustering method (von Luxburg, 2007), which is computationally fast. Compared to the adjacency matrix, which only has two values 0 and 1, the estimated edge probability matrix contains the probability value of each edge, which can provide more information about the relationship between nodes and better recover the graph structure. In addition, the estimated probability matrix is proven to have asymptotic normality, which can be used for uncertainty quantification and confidence interval estimation. By contrast, the adjacency matrix can not achieve this goal. We also establish a uniform error bound for the estimated edge probability matrix in terms of the entrywise maximum norm. These theoretical results are novel in the SBM literature. Furthermore, GNNs tend to majorly focus on information propagation among nodes and thus the model capability is limited, i.e., almost fail in learning topological and structural information. However, as recently shown by Wasserman (2018); Hensel et al. (2021); Pun et al. (2022); Liang et al. (2025); Goel et al. (2025); Dixon et al. (2025), such topological structures, e.g., connected components and holes might be an important step in graph knowledge discovery. For instance, persistent homology (Edelsbrunner, 2013; Zomorodian & Carlsson, 2004) has been used to study the topological information encoded in the graph (Zhao & Wang, 2019; Chen et al., 2021a;b; Li et al., 2025; Coskunuzer et al., 2024; Chen et al., 2023a; Horn et al., 2021; Arafat et al., 2025). However, these ideas have never been yet applied in conjunction with knowledge representation learning.

Aiming to solve the above challenges, we turn to the idea of combining learned community information from using an SBM-based model and different types of topological features by using multiple descriptor functions that can generate more expressive node embedding. The SBM is a powerful tool to learn from graph-structured data, as it is designed to model graphs with clear community structures. In GNNs, these communities represent groups of nodes having similar behaviors. By leveraging SBM in our topological neural networks (TNNs), the model can simultaneously learn the latent communities and capture the topological structures to enhance the performance in link prediction and node classification. Moreover, the community-based learning from SBM enables our TNNs to capture local structures to improve prediction accuracy, when dealing with sparse graphs with few connections, which in general is a great challenge for GNNs. SBM, on the other hand, provides a probabilistic framework that can help the model infer relationships between sparse connections as well as providing a better interpretability and understanding of the relationships between nodes. We use the learned community information from SBM together with different types of topological features by using multiple descriptor functions to generate more expressive latent node embedding. The primary contributions of this work can be summarized as follows:

- We propose a Stochastic Block Model-Aware Topological Neural Networks (SBM-TNN), a novel TNN-based model equipped with SBM concepts that captures topological structures, node features, and structure of neighborhood relations. It is the first approach bringing the concepts of topological signature representation learning and stochastic block models to graph learning.

- We further study the important problem, i.e., how to quantify the uncertainty for the estimated edge probability matrix. To achieve this goal, we establish an entrywise error bound and asymptotic normality of the estimated edge probability matrix that can be used to construct asymptotically valid confidence intervals for the edge probabilities, and help quantify the accuracy and uncertainty of the estimated edge probability matrix and provide theoretical guarantee for the follow up procedures of the proposed SBM-TNN model.

- Extensive experiments on benchmark datasets clearly show that SBM-TNN delivers state-of-the-art link prediction and knowledge graph completion tasks with a significant margin.

## 2 Related Work

**Graph Neural Networks.** Recently, Graph Neural Network (GNN) has emerged as a primary tool for node classification, link prediction, graph classification, and graph forecasting (Wu et al., 2020; Zhou et al., 2020; Chen & Gel, 2023; Chen et al., 2022c; Zhao et al., 2023; Chen et al., 2022a; 2023b; 2024b;a; Chen & Gel, 2025). Based on the spectral graph theory, Bruna et al. (2014) introduces a graph-based convolution in the Fourier domain. However, the complexity of this model is very high since all Laplacian eigenvectors are needed. To tackle this problem, ChebNet (Defferrard et al., 2016) integrates spectral graph convolution with Chebyshev polynomials. Then, Graph Convolutional Networks (GCNs) (Kipf & Welling, 2017) simplifies the graph convolution with a localized first-order approximation. SEAL (Zhang & Chen, 2018) extracts local enclosing subgraphs around the target links and learns a function mapping the subgraph patterns to link existence. Graph2Gauss (G2G) (Bojchevski & Günnemann, 2018) designs an unsupervised model that handles inductive link prediction by using a deep encoder to embed each node as a Gaussian distribution. Deep Generative Latent Feature Relational Model (DGLFRM) (Mehta et al., 2019) proposes an overlapping stochastic blockmodel for community discovery tasks. In addition, the Hyperbolic Graph Convolutional Neural Networks (HGCN) (Chami et al., 2019) leverages both the hyperbolic geometry and GCN framework to learn node representations. Another interesting recent strategy is to use pairwise topological features to find latent representations of the geometrical structure of graph using GCN (Yan et al., 2021). Moreover, Bi-Level Attention Graph Neural Networks (BA-GNN) (Iyer et al., 2021) leverages both node-node and relation-relation interactions (without meta paths) to capture more information about graph components. Mixed-Curvature Multi-Relational Graph Neural Network ($M^2$GNN) (Wang et al., 2021a) is designed to embed multi-relational information in a mixed-curvature space for knowledge graph (KG) completion. Additionally, Dual-Geometric Space Embedding Model (DGS) (Iyer et al., 2022) studies KG in complex non-Euclidean geometric space by modeling different views via different geometric spaces. A common limitation is that they fail to accurately capture correlated and rich topological properties of graphs and incorporate rich structure and topological information both in local and global domains.

**Stochastic Block Model.** The SBM (Holland et al., 1983) is a probabilistic model to generate networks with community structures, where nodes are partitioned into blocks and the probability of edges between nodes depends on their block memberships. The past few decades have seen various methods for recovering community memberships based on the observed network (Abbe, 2018). Spectral clustering stands out because of its computational tractability. The statistical properties of spectral clustering under SBM or its variant DCSBM have been widely studied. For example, the weak consistency of clustering, i.e., the proportion of misclustered nodes converges to zero as the number of nodes increases, has been investigated by Rohe et al. (2011); Lei & Rinaldo (2015); Joseph & Yu (2016), among others, and the strong consistency, namely, the memberships can be perfectly recovered in large samples, has been established in Su et al. (2019). Moreover, the minimax rate of the estimator for the edge probability matrix (i.e., the population counterpart of the adjacency matrix in terms of matrix norms such as Frobenius or spectral norm has been provided (Gao et al., 2015). However, how accurately the spectral clustering method can estimate each entry of the edge probability matrix is unclear. We develop the entrywise bound for the estimated edge probability matrix, which is essential to quantify the uncertainty in statistical inference for the population counterpart of the adjacency matrix in SBMs and DCSBMs. In addition, we establish the asymptotic normality of the estimated edge probability matrix, which turns out to be asymptotic efficient, under SBMs and DCSBMs, so we can construct asymptotically entrywise confidence intervals for the probability matrix. The asymptotic Gaussian behavior for the estimators of the block probability matrices (Tang et al., 2022) and the eigenvector matrix (Tang & Priebe, 2018; Cape et al., 2019; Xie, 2024) have been studied under SBMs. However, the asymptotic behavior of the estimator for the edge probability matrix under more general DCSBMs is undeveloped.

## 3 Stochastic Block Model-Aware Topological Neural Networks: Undirected Graph

### 3.1 Mixed-Up Undirected Graph Construction

To capture the topological information from the graph $\mathcal{G}$ and node features, we construct a mixed-up graph $\mathcal{G}_\mathcal{M} = (\boldsymbol{A}_\mathcal{M}, \boldsymbol{X})$ based on original input graph $\mathcal{G}_\mathcal{O} = (\boldsymbol{A}_\mathcal{O}, \boldsymbol{X})$ and $k$-hop graph $\mathcal{G}_\mathcal{K} = (\boldsymbol{A}_\mathcal{K}, \boldsymbol{X})$ (where $\boldsymbol{A}_\mathcal{M}$,

$\boldsymbol{A}_{\mathcal{O}}$, and $\boldsymbol{A}_{\mathcal{K}}$ denote adjacency matrices of the mixed-up graph, original graph, and $k$-hop graph respectively). $\boldsymbol{X} = [\boldsymbol{x}_1^\top, \boldsymbol{x}_2^\top, \ldots, \boldsymbol{x}_N^\top] \in \mathbb{R}^{N \times d}$ is the node feature matrix where $N$ is the number of nodes, $d$ is the number of features, and $\boldsymbol{x}_i$ denotes the node features of the node $u_i$.

**Original Graph Representation Learning.** We adopt the Graph Convolutional Layer (GCL) to perform message passing on the original graph $\mathcal{G}_{\mathcal{O}} = (\boldsymbol{A}_{\mathcal{O}}, \boldsymbol{X})$ where $\boldsymbol{A}_{\mathcal{O}}$ denotes the adjacency matrix of the original graph. It utilizes the original graph structure of $\mathcal{G}_{\mathcal{O}}$ with its node feature matrix $\boldsymbol{X}$ through the graph convolution operation and a multi-layer perceptron (MLP). Specifically, the designed graph convolution operation proceeds by multiplying the input of each layer with the $\tau$-th power of the normalized adjacency matrix. The $\tau$-th power operator contains statistics from the $\tau$-th step of a random walk on the graph (in this study, we set $\tau$ to be 2), thus nodes can indirectly receive more information from farther nodes in the graph. Combined with a multi-layer perceptron (MLP), the representation learned at the $\ell$-th layer is given by:

$$\boldsymbol{\mathcal{Z}}_{\mathcal{G}_{\mathcal{O}}}^{(\ell+1)} = f_{\text{MLP}}(\sigma(\hat{\boldsymbol{A}_{\mathcal{O}}}^\tau \boldsymbol{H}_{\mathcal{G}_{\mathcal{O}}}^{(\ell)} \boldsymbol{W}^{(\ell)})), \tag{1}$$

where $\hat{\boldsymbol{A}}_{\mathcal{O}} = \tilde{\boldsymbol{D}}_{\mathcal{O}}^{-\frac{1}{2}} \tilde{\boldsymbol{A}}_{\mathcal{O}} \tilde{\boldsymbol{D}}_{\mathcal{O}}^{\frac{1}{2}}$, $\tilde{\boldsymbol{A}}_{\mathcal{O}} = \boldsymbol{A}_{\mathcal{O}} + \boldsymbol{I}$, and $\tilde{\boldsymbol{D}}$ is the corresponding degree matrix of $\tilde{\boldsymbol{A}}$, $\boldsymbol{H}_{\mathcal{G}_{\mathcal{O}}}^{(0)} = \boldsymbol{X}$, $f_{\text{MLP}}$ is an MLP which has 2 layers with batch normalization, $\sigma(\cdot)$ is the non-linear activation function, $\boldsymbol{W}^{(\ell)}$ is a trainable weight of $\ell$-th layer.

$\mathcal{K}$**-Nearest Neighbor Graph Representation Learning.** First, in order to capture graph structural information of nodes in topology and feature spaces, we build a $\mathcal{K}$-nearest neighbor ($\mathcal{K}$NN) graph, i.e., $\mathcal{G}_{\mathcal{K}} = (\boldsymbol{A}_{\mathcal{K}}, \boldsymbol{X})$. In our study, we first define the similarity matrix $\boldsymbol{S}_{\mathcal{K}} \in \mathbb{R}^{N \times N}$ among $N$ nodes and we consider three different methods as follows: *(i) Cosine Similarity*: It uses the cosine value of the angle between two vectors to measure the similarity, i.e., $\boldsymbol{S}_{uv} = \frac{\boldsymbol{x}_u \cdot \boldsymbol{x}_v}{|\boldsymbol{x}_u||\boldsymbol{x}_v|}$; *(ii) Gaussian Kernel*: It is based on the idea of the heat equation, a partial differential equation that describes how heat propagates over time $t$, which can be defined as follows $\boldsymbol{S}_{uv} = \exp(-||\boldsymbol{x}_u - \boldsymbol{x}_v||^2/t)$; and *(iii) Node Embedding Similarity*: Let $\boldsymbol{H}^{(\ell+1)}$ be the node embedding of $(\ell)$-th layer of GNN. For any $u, v \in \mathcal{V}$, we can calculate the similarity score $\boldsymbol{S}_{uv}$ between nodes $u$ and $v$ as (i) Cosine Similarity: $\boldsymbol{S}_{uv} = \frac{\boldsymbol{H}_u^{(\ell+1)} \cdot \boldsymbol{H}_v^{(\ell+1)}}{||\boldsymbol{H}_u^{(\ell+1)}||||\boldsymbol{H}_v^{(\ell+1)}||}$ or (ii) Gaussian Kernel: $\boldsymbol{S}_{uv} = \exp\left(-||\boldsymbol{H}_u^{(\ell+1)} - \boldsymbol{H}_v^{(\ell+1)}||^2/t\right)$ (where $t$ is a free parameter). Then, the adjacency matrix $\boldsymbol{A}_{\mathcal{K}}$ can be obtained by selecting top-$\mathcal{K}$ similar neighboring nodes of each node. Similarly, we can use Eq. 1 to learn the $(\ell+1)$-th layer node embeddings of the above $\mathcal{K}$NN graph, which is denoted by $\boldsymbol{\mathcal{Z}}_{\mathcal{G}_{\mathcal{K}}}^{(\ell+1)}$.

**Mixup for Graph Construction.** Here we adopt the node-level attention mechanism to learn the hidden connectivity between nodes. Specifically, given a node pair $(u, v)$, the importance coefficient between nodes $u$ and $v$ can be formulated as (for the simplicity, we omit $(\ell+1)$ for $\boldsymbol{\mathcal{Z}}_{\mathcal{G}_{\mathcal{O}}}^{(\ell+1)}$ and $\boldsymbol{\mathcal{Z}}_{\mathcal{G}_{\mathcal{K}}}^{(\ell+1)}$):

$$\boldsymbol{e}_{uv}^{\mathcal{M}} = \boldsymbol{W}_{\mathcal{M}}[\boldsymbol{\mathcal{Z}}_{\mathcal{G}_{\mathcal{O}}}, \boldsymbol{\mathcal{Z}}_{\mathcal{G}_{\mathcal{K}}}],$$

$$\alpha_{e_{uv}^{\mathcal{M}}} = \text{Softmax}(\boldsymbol{e}_{uv}^{\mathcal{M}}) = \frac{\exp(\sigma(\boldsymbol{W}'_{\mathcal{M}} \boldsymbol{e}_{uv}^{\mathcal{M}}))}{\sum_{v' \in \mathcal{V}} \exp(\sigma(\boldsymbol{W}'_{\mathcal{M}} \boldsymbol{e}_{uv'}^{\mathcal{M}}))},$$

where $[\cdot, \cdot]$ represents the concatenation operation, $\boldsymbol{W}_{\mathcal{M}}$ and $\boldsymbol{W}'_{\mathcal{M}}$ are trainable parameters, $\sigma(\cdot)$ denotes the LeakyReLU function with negative input slope as 0.1. After the above calculation, we can get the mixup attention score $\alpha_{e_{uv}^{\mathcal{M}}}$ which represents the weight of the edge between nodes $u$ and $v$.

## 3.2 Stochastic Block Models for Undirected Graph

We consider two classes of probabilistic models for generating undirected networks with communities. The first is the SBM (Holland et al., 1983). The second is the DCSBM (Karrer & Newman, 2011). Suppose the $N$ nodes are assigned to $K$ non-overlapping communities. The $k$-th community has $N_k$ Nodes with $\sum_{k=1}^{K} N_k = N$, and denote $\pi_k := N_k/N$. Let $g_i \in \{1, ..., K\}$ be the community assignment (i.e., cluster) of node $i$. Alternatively, the community assignments can be represented by a membership matrix $\boldsymbol{Z} \in \{0, 1\}^{N \times K}$, where each row corresponds to a node and each column to a community. Specifically, $\boldsymbol{Z}_{ik} = 1$ if and only if $g_i = k$, and $\boldsymbol{Z}_{ij} = 0$ otherwise. Let $\boldsymbol{B} \in \mathbb{R}^{K \times K}$ be the block probability matrix, where $B_{st}$ specifies the probability of an edge between nodes in communities $s$ and $t$.

Given $\boldsymbol{B}$ and $\boldsymbol{Z}$, the SBM assume that each entry $\boldsymbol{A}_{ij}(i < j)$ of $\boldsymbol{A}$ is generated independently by $\boldsymbol{A}_{ij} \sim$ Bernoulli$(\boldsymbol{B}_{g_i g_j})$. In SBMs, the nodes within each community are stochastic equivalent. To incorporate the node heterogeneity, the more general model DCSBM is considered as follows. Let $\boldsymbol{\theta} = (\boldsymbol{\theta}_1, ..., \boldsymbol{\theta}_N)^{\mathsf{T}} \in \mathbb{R}^N$ be the node propensity parameters and denote $\boldsymbol{\Theta} = \text{diag}\{\boldsymbol{\theta}_1, ..., \boldsymbol{\theta}_N\}$, where each entry $\boldsymbol{\theta}_i$ captures the individual tendency or activity level of node $i$ to form edges in the network. Given $\boldsymbol{B}$, $\boldsymbol{Z}$ and $\boldsymbol{\Theta}$, the DCSBM assume each entry $\boldsymbol{A}_{ij}(i < j)$ of $\boldsymbol{A}$ is generated independently by $\boldsymbol{A}_{ij} \sim$ Bernoulli$(\boldsymbol{\theta}_i \boldsymbol{\theta}_j \boldsymbol{B}_{g_i g_j})$. It is then easy to see that

$$\boldsymbol{P} := \boldsymbol{\Theta Z B Z^{\mathsf{T}} \Theta} \in \mathbb{R}^{N \times N} \tag{2}$$

is the population counterpart of $\boldsymbol{A}$. $\boldsymbol{P}$ is referred to as the edge probability matrix. Note that $\boldsymbol{\Theta}$ and $\boldsymbol{B}$ are only identifiable up to scaling. As a remedy, we use the following normalization rule

$$\sum_{i,g_i=k} \boldsymbol{\theta}_i = N_k, \quad k = 1, ..., K.$$

With this normalization rule, the SBM is nested by the DCSBM by letting $\boldsymbol{\theta}_i = 1$ for $i = 1, ..., N$. To estimate $\boldsymbol{P}$, we should estimate $\boldsymbol{Z}$, $\boldsymbol{B}$ and $\boldsymbol{\Theta}$, respectively. Before that, we first recall and introduce some notation. Let $\hat{d}_i = \sum_{j=1}^{N} \boldsymbol{A}_{ij}$ be the degree of node $i$ and $\boldsymbol{D} = \text{diag}\{\hat{d}_1, ..., \hat{d}_N\}$. The graph Laplacian is defined as $\boldsymbol{L} = \boldsymbol{I} + \boldsymbol{D}^{-1/2} \boldsymbol{A} \boldsymbol{D}^{-1/2}$. Note that this graph Laplacian matrix has the same eigenspaces as that of $\boldsymbol{D}^{-1/2} \boldsymbol{A} \boldsymbol{D}^{-1/2}$.

Under SBMs, we estimate $\boldsymbol{Z}$ using the standard spectral clustering on the graph Laplacian matrix $\boldsymbol{L}$. That is, conducting $k$-means on the top-$K$ eigenvectors of $\boldsymbol{L}$. The estimator is denoted by $\hat{\boldsymbol{Z}}$. WLOG, we assume that $\hat{\boldsymbol{Z}}$ has been orthogonally transformed to align with $\boldsymbol{Z}$. We estimate $\boldsymbol{B}$ by the following $\hat{\boldsymbol{B}} = (\hat{\boldsymbol{B}}_{ql})_{1 \leq q \leq l \leq K}$,

$$\hat{\boldsymbol{B}}_{ql, q \neq l} := \frac{\sum_{\hat{g}_i=q, \hat{g}_j=l} \boldsymbol{A}_{ij}}{\hat{N}_q \hat{N}_l} \quad \text{and} \quad \hat{\boldsymbol{B}}_{qq} := \frac{\sum_{\hat{g}_i=q, \hat{g}_j=q} \boldsymbol{A}_{ij}}{\hat{N}_q (\hat{N}_q - 1)}.$$

Thereby, we obtain $\hat{\boldsymbol{P}} := \hat{\boldsymbol{Z}} \hat{\boldsymbol{B}} \hat{\boldsymbol{Z}}^{\mathsf{T}}$. Under the DCSBMs, we estimate $\boldsymbol{Z}$ using the spherical spectral clustering on the graph Laplacian matrix $\boldsymbol{L}$. That is, conducting $k$-means on the $L_2$-row-normalized top-$K$ eigenvectors of $\boldsymbol{L}$. With a light abuse of notation, the estimator is also denoted by $\hat{\boldsymbol{Z}}$. We estimate $\boldsymbol{B}$ by $\hat{\boldsymbol{B}} = (\hat{\boldsymbol{B}}_{ql})_{1 \leq q, l \leq K}$,

$$\hat{\boldsymbol{B}}_{ql} := \frac{\sum_{1 \leq i \neq j \leq N} \boldsymbol{A}_{ij} \hat{\boldsymbol{Z}}_{iq} \hat{\boldsymbol{Z}}_{jl}}{\sum_{1 \leq i \neq j \leq N} \hat{\boldsymbol{Z}}_{iq} \hat{\boldsymbol{Z}}_{jl}} = \frac{\sum_{\hat{g}_i=q, \hat{g}_j=l} \boldsymbol{A}_{ij}}{\hat{N}_q \hat{N}_l}.$$

We estimate $\boldsymbol{\theta}_i$ by $\hat{\boldsymbol{\theta}}_i$ defined as

$$\hat{\boldsymbol{\theta}}_i = \frac{\hat{N}_{\hat{g}_i} \sum_j \boldsymbol{A}_{ij}}{\sum_{\hat{g}_l=\hat{g}_i} \sum_{j=1}^{N} \boldsymbol{A}_{lj}},$$

where $\hat{N}_{\hat{g}_i}$ is the number of nodes in the estimated community $\hat{g}_i$. We also denote $\hat{\boldsymbol{\Theta}} = \text{diag}(\hat{\boldsymbol{\theta}})$. Finally, we obtain $\hat{\boldsymbol{P}} := \hat{\boldsymbol{\Theta}} \hat{\boldsymbol{Z}} \hat{\boldsymbol{B}} \hat{\boldsymbol{Z}}^{\mathsf{T}} \hat{\boldsymbol{\Theta}}$. The detailed derivations of the estimators can be found in the Appendix.

### 3.3 Multi-View Topological Graph Neural Networks for Undirected Graph

**Multi-View Topological Convolutional Layer.** To capture the underlying topological features of the subgraph $\mathcal{G}_u$ of each node $u$, we employ $\mathcal{K}$ filtration functions: $f_i : \mathcal{V} \mapsto \mathbb{R}$ for $i = \{1, ..., \mathcal{K}\}$. Each filtration function $f_i$ gradually reveals one specific topological structure at different levels of connectivity, e.g., degree centrality score, betweenness centrality score, closeness centrality score, and other node centrality measurements. With each filtration function $f_i$, we construct a set of $Q$ persistence images of resolution $P \times P$ using tools in persistent homology analysis. Combining $Q$ persistence images of resolution $P \times P$ from $\mathcal{K}$ different filtration functions, we construct a *multi-view* topological representation, i.e., the set of persistence images (PIs) $[\text{PI}_1, \text{PI}_2, ..., \text{PI}_{\mathcal{K}}]$ with the dimension $\mathcal{K} \times Q \times P \times P$. We design a multi-view topological convolutional layer $f_{\text{MV-GCL}}$ to (i) jointly extract and learn the latent topological features and (ii)

leverage and preserve the multi-modal structure. Firstly, hidden representations of the set of PIs are achieved through a combination of a CNN-based model and global pooling, which can be defined as

$$\boldsymbol{\mathcal{Z}}_{u,\mathcal{T}} = \xi_{\text{POOL}}(f_{\text{CNN}}([\text{PI}_1, \text{PI}_2, \ldots, \text{PI}_{\mathcal{K}}])), \tag{3}$$

where $f_{\text{CNN}}$ is a CNN-based neural network, $\xi_{\text{POOL}}$ is a pooling layer that preserves the information of the input in a fixed-size representation (in general, we consider either global average pooling or global max pooling). In Eq. equation 3, we first apply a CNN-based model to learn the latent feature of PIs, and then employ a global pooling layer over the latent feature and obtain an image-level feature.

**Graph Convolutional Layer.** Our third representation learning module is the Graph Convolutional Layer (GCL). It utilizes the graph structure of $\mathcal{G}_{\mathcal{M}}$ with its node feature matrix $\boldsymbol{X}$ through the graph convolution operation and a multi-layer perceptron (MLP). The representation learned at the $\ell$-th layer is given by

$$\boldsymbol{\mathcal{Z}}_{\mathcal{G}}^{(\ell+1)} = f_{\text{MLP}}([\sigma(\boldsymbol{A}_{\mathcal{M}}\boldsymbol{H}_{\mathcal{M}}^{(\ell)}\boldsymbol{W}_{\mathcal{M}}^{(\ell)}), \sigma(\hat{\boldsymbol{P}}\boldsymbol{H}_{\text{SBM}}^{(\ell)}\boldsymbol{W}_{\text{SBM}}^{(\ell)})]),$$

where $\boldsymbol{H}_{\mathcal{M}}^{(0)} = \boldsymbol{H}_{\text{SBM}}^{(0)} = \boldsymbol{X}$, $f_{\text{MLP}}$ is an MLP which has 2 layers with batch normalization, $\sigma(\cdot)$ is the non-linear activation function, $\boldsymbol{W}_{\mathcal{M}}^{(\ell)}$ and $\boldsymbol{W}_{\text{SBM}}^{(\ell)}$ are trainable weight matrices of $\ell$-th layer. Then, we obtain the final embedding $\boldsymbol{\mathcal{Z}}$ by combining embeddings from the above modules, i.e., $\boldsymbol{\mathcal{Z}} = [\boldsymbol{\mathcal{Z}}_{\mathcal{T}}, \boldsymbol{\mathcal{Z}}_{\mathcal{G}}]$, where $[\cdot, \cdot]$ denotes the concatenation operation and $\boldsymbol{\mathcal{Z}}_{\mathcal{G}}$ represents the final output of the graph convolutional layer.

# 4 Stochastic Block Model-Aware Topological Neural Networks: Knowledge Graph

A knowledge graph (KG) is defined as a directed graph that stores structured information about real-world entities and relations. Let $\mathcal{G} = \{\mathcal{V}, \mathcal{R}, \mathcal{L}\}$ be an instance of a KG, where $\mathcal{V}$, $\mathcal{R}$, and $\mathcal{L}$ denote the entity (i.e., node), relation, and edge sets respectively. Each edge $e \in \mathcal{L}$ is presented as a triple $(h, r, t) \in \mathcal{V} \times \mathcal{R} \times \mathcal{V}$, describing that there is a relationship $r \in \mathcal{R}$ from head entity $h$ to tail entity $t$. In order to apply the stochastic co-block model (which is introduced in Section 4.1) to classify nodes into $K$ clusters, we first transform the KG into a directed graph and then generate community information based on the graph's adjacency matrix instead of the knowledge graph by using SBM method. Note that, for the KG, we incorporate its information into the model and we don't have an additional assumption for KG. The overall architecture of SBM-TNN is as shown in Figure 1.

## 4.1 Stochastic Co-Block Models for Directed Graph

Similar to the undirected networks, we consider two classes of probabilistic models for generating directed networks with co-clusters, namely, row clusters (communities) and column clusters (communities). The first is the stochastic co-block model (ScBM), and the second is the degree-corrected stochastic co-block model (DCScBM) (Rohe et al., 2016).

Different from SBMs and DCSBMs, the models for directed networks incorporate two kinds of clusters. Suppose the $N$ nodes are assigned to $K^y$ non-overlapping row clusters and $K^z$ non-overlapping column clusters. WLOG, suppose $K^y \leq K^z$. The $k^y$th (resp. $k^z$th) row (resp. column) cluster has $N_k^y$ (resp. $N_k^z$) nodes with $\sum_{k=1}^{K^y} N_k^y = N$ (resp. $\sum_{k=1}^{K^z} N_k^z = N$), and denote $\pi_k^y := N_k^y/N$ (resp. $\pi_k^z := N_k^z/N$). Let $g_i^y \in \{1, ..., K^y\}$ (resp. $g_i^z \in \{1, ..., K^z\}$) be the row (resp. column) community assignment of node $i$. Following the same logic as undirected networks, let $\boldsymbol{Y} \in \{0,1\}^{N \times K^y}$ and $\boldsymbol{Z} \in \{0,1\}^{N \times K^z}$ denote the row and column membership matrices, respectively. Let $\boldsymbol{B} \in \mathbb{R}^{K^y \times K^z}$ be the block probability matrix.

Given $\boldsymbol{B}$, $\boldsymbol{Y}$ and $\boldsymbol{Z}$, the ScBM assume that each entry $\boldsymbol{A}_{ij}(i \neq j)$ of $\boldsymbol{A}$ is generated independently by $\boldsymbol{A}_{ij} \sim \text{Bernoulli}(\boldsymbol{B}_{g_i^y g_j^z})$. In ScBMs, the nodes in a common row (resp. column) cluster are stochastically equivalent senders (resp. receivers) in the sense that they send (resp. receive) out an edge to a third node with equal probabilities. To incorporate the node heterogeneity in sending and receiving edges, the more general model DCScBM is considered as follows. Let $\boldsymbol{\theta}^y = (\boldsymbol{\theta}_1^y, ..., \boldsymbol{\theta}_N^y)^{\mathsf{T}} \in \mathbb{R}^N$ and $\boldsymbol{\theta}^z = (\boldsymbol{\theta}_1^z, ..., \boldsymbol{\theta}_N^z)^{\mathsf{T}} \in \mathbb{R}^N$ be the node propensity parameters in sending and receiving edges, respectively. Denote $\boldsymbol{\Theta}^y = \text{diag}(\boldsymbol{\theta}^y)$ and

$\mathbf{\Theta}^z = \text{diag}(\boldsymbol{\theta}^z)$. Given $\boldsymbol{B}, \boldsymbol{Y}, \boldsymbol{Z}, \mathbf{\Theta}^y$ and $\mathbf{\Theta}^z$, the DCScBM assumes each entry $\boldsymbol{A}_{ij}(i \neq j)$ of $\boldsymbol{A}$ is generated independently by $\boldsymbol{A}_{ij} \sim \text{Bernoulli}(\boldsymbol{\theta}_i^y \boldsymbol{\theta}_j^z \boldsymbol{B}_{g_i^y g_j^z})$. Then, under the DCScBM, it is easy to see

$$\boldsymbol{P} := \mathbf{\Theta}^y \boldsymbol{Y} \boldsymbol{B} \boldsymbol{Z}^\intercal \mathbf{\Theta}^z \in \mathbb{R}^{N \times N} \tag{4}$$

is the population adjacency matrix of $\boldsymbol{A}$, referred to as the edge probability matrix later. To ensure identifiability, we use the following normalization rule

$$\begin{aligned} \sum_{i, g_i^y = k^y} \boldsymbol{\theta}_i^y &= N_k^y, \quad k^y = 1, ..., K^y, \\ \sum_{i, g_i^z = k^z} \boldsymbol{\theta}_i^z &= N_k^z, \quad k^z = 1, ..., K^z. \end{aligned} \tag{5}$$

The ScBM is nested by the DCScBM by letting $\boldsymbol{\theta}_i^y = 1$ and $\boldsymbol{\theta}_i^z = 1$ for $i = 1, ..., N$. To estimate $\boldsymbol{P}$, we introduce some notation now. Let $\hat{d}_i^y = \sum_{j=1}^N \boldsymbol{A}_{ij}$ (resp. $\hat{d}_i^z = \sum_{j=1}^N \boldsymbol{A}_{ji}$) be the out-degree (resp. in-degree) of node $i$ and $\boldsymbol{D}^y = \text{diag}\{\hat{d}_1^y, ..., \hat{d}_N^y\}$ (resp. $\boldsymbol{D}^z = \text{diag}\{\hat{d}_1^z, ..., \hat{d}_N^z\}$). Define the graph Laplacian by $\boldsymbol{L} = \boldsymbol{I} + (\boldsymbol{D}^y)^{-1/2} \boldsymbol{A} (\boldsymbol{D}^z)^{-1/2}$. Under ScBMs, we compute the SVD of $\boldsymbol{L}$ and then conduct the $k$-means on the top-$K^y$ left (resp. top-$K^z$ right) singular vectors of $\boldsymbol{L}$, to obtain $K^y$ row (resp. $K^z$ column) clusters, denoted by $\hat{\boldsymbol{Y}}$ (resp. $\hat{\boldsymbol{Z}}$). WLOG, we assume that $\hat{\boldsymbol{Y}}$ ($\hat{\boldsymbol{Z}}$) has been orthogonally transformed to align with $\boldsymbol{Y}$ (resp. $\boldsymbol{Z}$). Similar to the undirected set-up, we estimate $\boldsymbol{B}$ by the following $\hat{\boldsymbol{B}} = (\hat{\boldsymbol{B}}_{ql})_{1 \leq q \leq K^y, 1 \leq l \leq K^z}$,

$$\hat{\boldsymbol{B}}_{ql} := \frac{\sum_{1 \leq i \neq j \leq N} \boldsymbol{A}_{ij} \hat{\boldsymbol{Y}}_{iq} \hat{\boldsymbol{Z}}_{jl}}{\sum_{1 \leq i \neq j \leq N} \hat{\boldsymbol{Y}}_{iq} \hat{\boldsymbol{Z}}_{jl}} = \frac{\sum_{\hat{g}_i^y = q, \hat{g}_j^z = l} \boldsymbol{A}_{ij}}{\hat{N}_q^y \hat{N}_l^z},$$

where $\hat{N}_q^y$ (resp. $\hat{N}_l^z$) denotes the number of nodes in the estimated row cluster $q$ (resp. column cluster $l$). Thereby, we obtain $\hat{\boldsymbol{P}} := \hat{\boldsymbol{Y}} \hat{\boldsymbol{B}} \hat{\boldsymbol{Z}}^\intercal$. Under the DCScBMs, we estimate $\boldsymbol{Z}$ by applying the spherical spectral clustering to the graph Laplacian matrix $\boldsymbol{L}$. That is, we obtain the top-$K^y$ row and column singular vectors by computing the SVD of $\boldsymbol{L}$, and then conduct $k$-means on the $L_2$-row-normalized left and right singular vectors to obtain the $K^y$ row clusters and $K^z$ column clusters, respectively. With a light abuse of notation, the estimators are denoted by $\hat{\boldsymbol{Y}}$ and $\hat{\boldsymbol{Z}}$. Following the same logic as in DCSBM, we obtain the estimators $\hat{\boldsymbol{B}} = (\hat{\boldsymbol{B}}_{ql})$, $\hat{\mathbf{\Theta}}^y = \text{diag}(\hat{\boldsymbol{\theta}}^y)$ and $\hat{\mathbf{\Theta}}^z = \text{diag}(\hat{\boldsymbol{\theta}}^z)$ as follows.

$$\begin{aligned} \hat{\boldsymbol{B}}_{ql} &:= \frac{\sum_{\hat{g}_i^y = q, \hat{g}_j^z = l} \boldsymbol{A}_{ij}}{\hat{N}_q^y \hat{N}_l^z}, \quad \hat{\boldsymbol{\theta}}_i^y = \frac{\hat{N}_{\hat{g}_i^y} \sum_j \boldsymbol{A}_{ij}}{\sum_{\hat{g}_l^y = \hat{g}_i^y} \sum_{j=1}^N \boldsymbol{A}_{lj}}, \\ \hat{\boldsymbol{\theta}}_j^z &= \frac{\hat{N}_{\hat{g}_j^z} \sum_i \boldsymbol{A}_{ij}}{\sum_{\hat{g}_l^z = \hat{g}_j^z} \sum_{i=1}^N \boldsymbol{A}_{il}}, \end{aligned}$$

where $\hat{N}_{\hat{g}_i^y}$ is the number of nodes in the estimated row cluster $\hat{g}_i^y$ and $\hat{N}_{\hat{g}_j^z}$ is the number of nodes in the estimated column cluster $\hat{g}_j^z$. With these estimators at hand, we obtain $\hat{\boldsymbol{P}} := \hat{\mathbf{\Theta}}^y \hat{\boldsymbol{Y}} \hat{\boldsymbol{B}} \hat{\boldsymbol{Z}}^\intercal \hat{\mathbf{\Theta}}^z$.

### 4.2 Attention-Based Topological Neural Networks for Knowledge Graph

In this section, we propose a novel attention-based topological neural network (A-TNN) to combine the community information and topological signatures learned from ScBM for entity representation learning.

$$\begin{aligned} \boldsymbol{Z}_h^{(\ell+1)} &= \sigma\Big( \sum_{(r,t) \in \mathcal{N}_h} \alpha_{h,r,t}^{(\ell)} \tilde{\boldsymbol{X}}_{h,r,t}^{(\ell)} \Big), \\ \tilde{\boldsymbol{X}}_{h,r,t}^{(\ell)} &= \boldsymbol{W}_{\text{KG}_1}[\boldsymbol{X}_h^{(\ell)}, \boldsymbol{X}_t^{(\ell)}, \boldsymbol{X}_{C_k}^{(\ell)}, \boldsymbol{X}_h^{\text{topo}}, \boldsymbol{X}_r], \\ \alpha_{h,r,t}^{(\ell)} &= \frac{\exp(\sigma(\boldsymbol{W}_{\text{KG}_2}^{(\ell)} \tilde{\boldsymbol{X}}_{h,r,t}^{(\ell)}))}{\sum_{(r,t') \in \mathcal{N}_h} \exp(\sigma(\boldsymbol{W}_{\text{KG}_2}^{(\ell)} \tilde{\boldsymbol{X}}_{h,r,t}^{(\ell)}))}, \end{aligned} \tag{6}$$

where $\boldsymbol{X}_h^{(\ell)}$, $\boldsymbol{X}_t^{(\ell)}$, and $\boldsymbol{X}_r^{(\ell)}$ denote embeddings of head entity $h$, tail entity $t$, and relation $r$ respectively, $\boldsymbol{W}_{\mathrm{KG}_1}$ and $\boldsymbol{W}_{\mathrm{KG}_2}$ denote the linear transformation matrices, $\boldsymbol{X}_{C_k}^{(\ell)}$, $\mathcal{N}_h$ denotes the set of neighboring tuples $(r, t)$ for entity $h$, $\boldsymbol{X}_k^{(\ell)} = \sum_{h \in C_k} \boldsymbol{X}_h^{(\ell)}$ which aggregates node embeddings from the community $C_k$ (i.e., results from Section 4.1), and $\boldsymbol{X}_h^{\mathrm{topo}} = f_{\mathrm{MV\text{-}GCL}}(\mathcal{G}_h)$ (where $\mathcal{G}_h$ denotes the subgraph of the node $h$).

## 5 Theoretical Guarantees

To establish the statistical property of $\hat{\boldsymbol{P}}$, we need the following assumptions. In order to provide the results uniformly and reduce redundancy, we use the notations in the more general ScBMs and DCScBMs. The SBMs and DCSBMs can be regarded as the special case of ScBMs and DCScBMs.

**Assumption 5.1.** *Suppose $K^y$ and $K^z$ are both fixed and both the row and column clusters satisfies that for any $k^y \in \{1, ..., K^y\}$, $cN/K^y \leq N_k^y \leq CN/K^y$ for some constants $0 \leq c \leq C$; for any $k^z \in \{1, ..., K^z\}$, $c'N/K^z \leq N_k^z \leq C'N/K^z$ for some constants $0 \leq c' \leq C'$.*

**Assumption 5.2.** *Suppose the edge probability matrix $\boldsymbol{B} \in [0, 1]^{K^y \times K^z}$ $(K^y \leq K^z)$ is of rank $K^z$. The entries of $\boldsymbol{B}$ are of the same magnitude $\rho_N$ with $N\rho_N \geq c \log N$.*

**Assumption 5.3.** *Define $\sigma_{K^y}$ be the $K^y$th singular value of $\mathcal{L} = (\mathcal{D}^y)^{-1/2} \boldsymbol{P}(\mathcal{D}^z)^{-1/2}$, where $\mathcal{D}^y = \mathrm{diag}\{d_1^y, ..., d_N^y\}$ with $d_i^y = \sum_{j=1}^N \boldsymbol{P}_{ij}$, and $\mathcal{D}^z = \mathrm{diag}\{d_1^z, ..., d_N^z\}$ with $d_i^z = \sum_{j=1}^N \boldsymbol{P}_{ji}$. Suppose $\liminf_N |\sigma_K^y| > 0$.*

**Assumption 5.4.** *Define $\overline{\boldsymbol{\theta}}^y = \max_i \boldsymbol{\theta}_i^y$ and $\underline{\boldsymbol{\theta}}^y = \min_i \boldsymbol{\theta}_i^y$. Suppose $c \leq \liminf_N \underline{\boldsymbol{\theta}}^y \leq \limsup_N \overline{\boldsymbol{\theta}}^y \leq C$ for some constants $0 \leq c \leq C$. Similarly, define $\overline{\boldsymbol{\theta}}^z = \max_i \boldsymbol{\theta}_i^z$ and $\underline{\boldsymbol{\theta}}^z = \min_i \boldsymbol{\theta}_i^z$. Suppose $c' \leq \liminf_N \underline{\boldsymbol{\theta}}^z \leq \limsup_N \overline{\boldsymbol{\theta}}^z \leq C'$ for some constants $0 \leq c' \leq C'$.*

**Assumption 5.5.** *Define $\boldsymbol{H} = (\boldsymbol{Y}^{\mathsf{T}} \boldsymbol{\Theta}^y \boldsymbol{Y})^{1/2} \boldsymbol{B}_L (\boldsymbol{Z}^{\mathsf{T}} \boldsymbol{\Theta}^z \boldsymbol{Z})^{1/2} \in \mathbb{R}^{K^y \times K^z}$ with $\boldsymbol{B}_L := \boldsymbol{O}_B^{-1/2} \boldsymbol{B} \boldsymbol{P}_B^{-1/2}$, where $\boldsymbol{O}_B$ is a $K^y \times K^y$ diagonal matrix with $[\boldsymbol{O}_B]_{ss} = \sum_t \boldsymbol{B}_{st} n_t^z$ and $\boldsymbol{P}_B$ is a $K^z \times K^z$ diagonal matrix with $[\boldsymbol{P}_B]_{tt} = \sum_s \boldsymbol{B}_{st} n_s^y$. Suppose there exits gap between any two columns of $\boldsymbol{H}$ such that $\min_{i \neq j} \|\boldsymbol{H}_{\cdot i} - \boldsymbol{H}_{\cdot j}\|_2 \geq \xi$ for some constant $\xi > 0$.*

Assumptions 5.1-5.4 are generally required for undirected network models SBMs and DCSBMs. For directed network models ScBMs and DCScBMs, we need additional Assumption 5.5. Assumption 5.1 requires that the number of nodes in each cluster is not too small, which ensures the strong consistency of the spectral method, namely, each node is correctly clustered. For notational simplicity, we fix the number of communities, although the framework can naturally be extended to accommodate varying numbers of communities as in Su et al. (2019). Assumption 5.2 requires that the network is not too sparse. This condition is the minimal requirement for strong consistency of SBM (Abbe et al., 2015; Su et al., 2019; Ma et al., 2021). Assumption 5.3 implies that the singular value of the population Laplacian matrix is lower bounded by a constant. For an SBM with cross-block probability being $r$, and within-block probability being $r + p$, then $\sigma_K = p/(Kr + p)$ is a constant provided that $r$ and $p$ are of the same order. Assumption 5.4 requires that the node propensity parameters are upper and lower bounded. The upper bound is mild as the node propensity parameters are normalized to satisfy (5). The lower bound can be relaxed to $n^{-\alpha}$ for some positive constant $\alpha$ with a sacrifice of the simplicity of other conditions (Su et al., 2019). This assumption is only needed when networks follow DCSBMs and DCScBMs. Assumption 5.5 is required to ensure that there exists a gap between two rows of the population right singular vectors when the two nodes are in different column clusters. This is a remedy assumption for the invalidity of the full column rank under directed network models.

**Lemma 5.6** (Strong consistency). *Suppose Assumptions 5.1-5.5 hold. Then for large enough $N$, it holds for both ScBMs and DCScBMs that*

$$\sup_{1 \leq i \leq N} \mathbf{1}\{\hat{g}_i^y \neq g_i^y\} = 0 \text{ and } \sup_{1 \leq i \leq N} \mathbf{1}\{\hat{g}_i^z \neq g_i^z\} = 0 \text{ a.s.} \tag{7}$$

*and*

$$\sup_{1 \leq q \leq K^y, 1 \leq l \leq K^z} |\hat{\boldsymbol{B}}_{ql} - \boldsymbol{B}_{ql}| = O_{a.s.}(\frac{\sqrt{\rho \log N}}{N}). \tag{8}$$

*For DCScBMs, it also holds that*

$$\sup_{1 \le i \le N} |\hat{\boldsymbol{\theta}}_i^y - \boldsymbol{\theta}_i^y| = O_{a.s.}(\frac{\log N}{N\rho})$$

$$\sup_{1 \le i \le N} |\hat{\boldsymbol{\theta}}_i^z - \boldsymbol{\theta}_i^z| = O_{a.s.}(\frac{\log N}{N\rho}). \tag{9}$$

**Remark 1.** *Lemma 5.6 also holds for SBMs and DCSBMs with the notation slightly changed and without the requirement of Assumption 5.5. The strong consistency results are critical for deriving the error bound and asymptotic normality of $\hat{\boldsymbol{P}}$. equation 7 and equation 9 can be implied by Corollary III.1 and Theorem III.6 in Su et al. Su et al. (2019).*

**Theorem 5.7** (Error bound)**.** *Suppose Assumptions 5.1-5.5 hold. Then for large enough $N$, it holds for ScBMs that*

$$\sup_{1 \le i,j \le N} |\hat{\boldsymbol{P}}_{ij} - \boldsymbol{P}_{ij}| = O_{a.s.}(\frac{\sqrt{\rho \log N}}{N}), \tag{10}$$

*and for DCScBMs that*

$$\sup_{1 \le i,j \le N} |\hat{\boldsymbol{P}}_{ij} - \boldsymbol{P}_{ij}| = O_{a.s.}(\frac{\log N}{N}). \tag{11}$$

**Remark 2.** *For SBMs and DCSBMs, equation 10 and equation 22 hold without Assumption 5.5. Most of the current literature on SBM and its variants study the high probability spectral norm bound of $\boldsymbol{A}$ from $\boldsymbol{P}$ Gao et al. (2015); Lei & Rinaldo (2015). Our results are novel in that we provide the uniform entrywise bound of $\hat{\boldsymbol{P}}$ from $\boldsymbol{P}$ which holds almost surely. The less tightness of DCSBMs and DCScBMs comes from the estimation of the node propensity parameters.*

**Theorem 5.8** (Asymptotic normality)**.** *Suppose Assumptions 5.1-5.5 hold. Then for ScBMs, (12) holds with $c_0 = 1, \boldsymbol{\theta}_i^y = \boldsymbol{\theta}_j^z = 1$ and $\boldsymbol{E} = \boldsymbol{0}$. For DCScBMs, (12) holds with $c_0 = 1$ and $\boldsymbol{E}_{ij} = O_{a.s.}(\log N/N)$ (where $1 \le i,j \le N$).*

$$N(\hat{\boldsymbol{P}} - \boldsymbol{P} + \boldsymbol{E})_{ij} \to N(0, c_0 \cdot (\boldsymbol{\theta}_i^y)^2 \cdot \frac{\boldsymbol{B}_{g_i^y g_j^z}(1 - \boldsymbol{B}_{g_i^y g_j^z})}{\pi_{g_i^y} \pi_{g_j^z}} \cdot (\boldsymbol{\theta}_j^z)^2), \tag{12}$$

**Remark 3.** *(12) also holds for SBMs and DCSBMs without Assumption 5.5 and with the same order of $\boldsymbol{E}$ and notation slightly modified except that $c_0 = 2$ when $g_i = g_j$ and $c_0 = 1$ when $g_i \ne g_j$. For SBMs, the estimator $\hat{\boldsymbol{P}}$ is asymptotic efficient (Bickel et al., 2013). For DCSBMs and DCScBMs, the bias $\boldsymbol{E}_{ij}$ and $\boldsymbol{F}_{ij}$ come from the estimation of $\boldsymbol{\theta}$. Note that for denser networks, namely, $\rho = \Omega(\log N/N)$, the bias is dominated by the signal $\boldsymbol{P}_{ij}$. To the best of our knowledge, this is the first result to show the asymptotic normality of an estimator against the edge probability matrix $\boldsymbol{P}$.*

# 6 Experiments

**Datasets and Baselines.** We experiment on 2 types of networks for link prediction (i) citation networks: Cora-ML, Citeseer, and PubMed (Sen et al., 2008) and (ii) graphs related to Amazon shopping records: Photo and Computers (Shchur et al., 2018). For link prediction tasks, we compare against 10 state-of-the-art (SOA) baselines, including (i) Graph convolution network (GCN) (Kipf & Welling, 2017); (ii) Graph Attention Networks (GAT) (Veličković et al., 2018b); (iii) Hyperbolic Graph Convolutional Neural Networks (HGCN) (Chami et al., 2019); (iv) Position-aware Graph Neural Networks (P-GNN) (You et al., 2019); (v) SEAL (Zhang & Chen, 2018); (vi) Block Simplicial Complex Neural Networks (BScNets) (Chen et al., 2022b); (vii) Topological Loop-Counting Graph Neural Network (TLC-GNN) (Yan et al., 2021). For knowledge graph completion tasks, we conduct experiments on 3 well-known KG datasets including (i) FB15k-237 (Toutanova et al., 2015; Toutanova & Chen, 2015), WN18RR Dettmers et al. (2018), and NELL-995 (Xiong et al., 2017), and use the following popular models as baselines: (i) CompGCN (Vashishth et al., 2019); (ii) Relational Graph Convolutional Network (RGCN) (Schlichtkrull et al., 2018); (iii) KBGAT (Nathani et al., 2019); (iv) Atrous Convolution and Residual Embedding (AcrE) (Ren et al., 2020); (v) ReInceptionE (Xie et al.,

Table 1: AUC-ROC score on different link prediction benchmarks.

| Model | Cora-ML | Citeseer | PubMed | Photo | Computers |
|---|---|---|---|---|---|
| GCN Kipf & Welling (2017) | 90.5±0.2 | 82.6±1.9 | 89.6±3.7 | 91.8±0.0 | 87.8±0.0 |
| GAT Veličković et al. (2018b) | 72.8±0.2 | 74.8±1.5 | 80.3±0.0 | 92.9±0.3 | 86.4±0.0 |
| HGCN Chami et al. (2019) | 93.8±0.1 | 96.6±0.1 | 96.3±0.0 | 95.4±0.0 | 93.6±0.0 |
| P-GNN You et al. (2019) | 74.1±2.4 | 73.9±2.6 | 79.6±0.5 | 90.9±0.7 | 88.3±1.0 |
| SEAL Zhang & Chen (2018) | 91.3±5.7 | 89.8±2.3 | 92.4±1.2 | 97.8±1.3 | 96.8±1.5 |
| BScNets Chen et al. (2022b) | 94.9±0.7 | 95.5±0.5 | 97.6±0.1 | 96.6±0.3 | 97.0±0.3 |
| TLC-GNN Yan et al. (2021) | 94.9±0.4 | 95.1±0.7 | 97.0±0.1 | 98.2±0.1 | 97.9±0.1 |
| **SBM-TNN (ours)** | **96.2±0.2** | **97.1±0.3** | **98.2±0.1** | **98.8±0.2** | **99.0±0.2** |

2020); (vi) Semantic Evidence aware Graph Neural Network (SE-GNN) (Li et al., 2022); and (vii) Neural Bellman-Ford Network (NBFNet) (Zhu et al., 2021).

**Experiment Settings.** We implement our proposed SBM-TNN with Pytorch framework on two NVIDIA RTX A5000 GPUs with 24 GiB RAM. Following Chami et al. (2019), for graph link prediction tasks, we randomly split edges into 85%/5%/10% for training, validation, and testing, and we evaluate link prediction using the ROC-AUC score on the test set. For KG completion tasks, we follow the settings in previous works (Vashishth et al., 2019; Schlichtkrull et al., 2018), i.e., triplets in these datasets are randomly split into training, validation, and test sets respectively, and we evaluate the KG completion performance by using Mean Reciprocal Rank (MRR) and Hits@$N$ (here we consider $N \in \{1, 3, 10\}$). Code and data are publicly available at `https://github.com/yuzhouguangc/SBM-TNN`. For further details on the experiment settings, please refer to Appendix.

**Experiment Results.** The link prediction and KG completion results are summarized in Tables 1 and 2. From Table 1, the results indicate that our SBM-TNN consistently achieves the best performance on all datasets. More specifically, we find that (i) Compared to the spectral-based GNN (i.e., GCN), our SBM-TNN yields up to 12.8% relative improvements for all 5 datasets; (ii) Compared to the spatial-based GNNs (i.e., GAT, P-GNN, and SEAL), SBM-TNN improves upon the runner-up by a margin of 5.4%, 8.1%, 6.3%, 1.0%, and 2.3% on datasets Cora-ML, Citeseer, PubMed, Photo and Computers; (iii) SBM-TNN outperforms the hyperbolic-based NNs, i.e., HGCN with a statistically significant margin; (iv) SBM-TNN further improves topology-based GNN (i.e., BScNets and TLC-GNN) with a significant margin on all 5 datasets. We also compare with two additional sate-of-the-art baselines, i.e., NCNC (Wang et al., 2024) and LPFormer (Shomer et al., 2024) on Cora-ML and Citeseer datasets with MRR, and we observe that our SBM-TNN achieves average improvements of 23.1% and 3.3% over NCNC and LPFormer respectively (further details can be found in the Appendix). Additionally, Table 2 shows the performance of SBM-TNN and baseline methods on 3 KG datasets. From Table 2, we observe that SBM-TNN surpasses the baselines in terms of the MRR, Hits@1, Hits@3, and Hits@10 on all datasets. Furthermore, we have conducted an additional comparison with the GNNs + NBFNet on ogbl-wikikg2 data (Hu et al., 2020). The test MRR of SBM-TNN and GNNs + NBFNet are 0.7121±0.0009 and 0.7086±0.0028, i.e., our SBM-TNN is significantly better than this state-of-the-art method. Overall, the results show that SBM-TNN can accurately capture and learn the key structural and local topological information, and achieve highly promising performance in both link prediction and KG completion tasks.

Table 2: KGC results (%) with different scoring functions.

| | FB15k-237 | | | | WN18RR | | | | NELL-995 | | | |
|---|---|---|---|---|---|---|---|---|---|---|---|---|
| | MRR | Hits@1 | Hits@3 | Hits@10 | MRR | Hits@1 | Hits@3 | Hits@10 | MRR | Hits@1 | Hits@3 | Hits@10 |
| CompGCN Vashishth et al. (2019) | 35.5±0.1 | 26.4±0.1 | 39.0±0.2 | 53.6±0.2 | 47.2±0.2 | 43.7±0.3 | 48.5±0.3 | 54.0±0.0 | 38.1±0.4 | 30.4±0.5 | 42.2±0.3 | 52.9±0.1 |
| RGCN Schlichtkrull et al. (2018) | 29.6±0.3 | 19.1±0.5 | 34.0±0.2 | 50.1±0.2 | 43.0±0.2 | 38.6±0.3 | 45.0±0.1 | 50.8±0.3 | 27.8±0.2 | 19.9±0.2 | 31.4±0.0 | 43.0±0.3 |
| KBGAT Nathani et al. (2019) | 35.0±0.3 | 26.0±0.3 | 38.5±0.3 | 53.1±0.3 | 46.4±0.2 | 42.6±0.2 | 47.9±0.3 | 53.9±0.2 | 37.4±0.6 | 29.7±0.7 | 41.4±0.8 | 52.0±0.4 |
| AcrE Ren et al. (2020) | 35.8±0.3 | 26.6±0.2 | 39.3±0.3 | 54.5±0.2 | 45.9±0.2 | 42.2±0.3 | 47.3±0.2 | 53.2±0.1 | - | - | - | - |
| ReInceptionE Xie et al. (2020) | 34.9±0.2 | - | - | 52.8±0.2 | 48.3±0.3 | - | - | 58.2±0.3 | - | - | - | - |
| SE-GNN Li et al. (2022) | 36.1±0.3 | 23.4±0.2 | 37.0±0.3 | 51.5±0.2 | 48.4±0.4 | 43.6±0.2 | 47.9±0.2 | 57.2±0.3 | 39.3±0.5 | 30.2±0.3 | 43.0±0.2 | 52.8±0.3 |
| GNNs + NBFNet Zhu et al. (2021) | 41.5±0.1 | 32.1±0.1 | 45.6±0.4 | 59.9±0.3 | 55.1±0.1 | 49.7±0.1 | 57.2±0.3 | 66.6±0.3 | 40.5±0.3 | 32.7±0.2 | 44.9±0.3 | 55.0±0.4 |
| **SBM-TNN (ours)** | 36.9±0.2 | 27.3±0.1 | 40.8±0.2 | 55.6±0.1 | 49.1±0.2 | 44.1±0.2 | 48.9±0.1 | 59.3±0.2 | 41.7±0.2 | 34.7±0.2 | 45.2±0.1 | 55.2±0.1 |
| **SBM-TNN + NBFNet (ours)** | **41.9±0.1** | **33.0±0.1** | **47.1±0.2** | **60.1±0.2** | **55.5±0.1** | **49.9±0.1** | **59.2±0.1** | **67.0±0.0** | **42.0±0.1** | **34.9±0.2** | **45.5±0.3** | **55.9±0.1** |

Table 3: Link prediction (AUC-ROC) of SBM-TNN with different $\tau$.

| Dataset | $\tau = 1$ | $\tau = 2$ | $\tau = 3$ | $\tau = 4$ |
|---|---|---|---|---|
| Cora-ML | 93.7±0.8 | **96.2±0.2** | 95.4±0.3 | 93.2±0.9 |
| Citeseer | 94.9±0.8 | 97.0±0.3 | **97.1±0.3** | 95.1±0.6 |
| PubMed | 97.9±0.3 | **98.2±0.2** | 98.7±0.6 | 95.5±0.7 |

Table 4: Performance comparison for link prediction (AUC-ROC) between $\boldsymbol{P}$ and $\hat{\boldsymbol{P}}$.

| Architecture | Cora-ML | Citeseer | PubMed |
|---|---|---|---|
| SBM-TNN ($\boldsymbol{P}$) | 95.3±0.3 | 95.0±0.5 | 96.3±0.3 |
| SBM-TNN ($\hat{\boldsymbol{P}}$) | **96.2±0.2** | **97.1±0.3** | **98.2±0.1** |

**Sensitivity Analysis** To evaluate the link prediction performance of SBM-TNN with different $\tau$, we conduct experiments on the Cora-ML, Citeseer, and PubMed datasets. As shown in Table 3, we observe that our approach achieves optimal performance under a specific power of the normalized adjacency matrix. In addition, we have run additional experiments on Cora-ML and Citeseer. As shown in Table 4, SBM-TNN with estimated $\hat{\boldsymbol{P}}$ outperforms SBM-TNN with actual $\boldsymbol{P}$. In particular, the average relative gain of SBM-TNN with estimated $\hat{\boldsymbol{P}}$ over SBM-TNN with actual $\boldsymbol{P}$ is 1.58%.

**Ablation Study** We have also conducted ablation studies to explore the importance of different components, and considered two ablated variants, i.e., (i) SBM-GNN represents that replacing TNN by a graph neural network (GNN), and (ii) TNN represents that SBM-TNN without adding SBM method, (i.e., we do not incorporate community-level information into the model architecture). From Table 5, we observe that our SBM-TNN always outperforms both the GNN model equipped with SBM (i.e., SBM-GNN) and TNN model on Cora-ML and Citeseer data. That is, when ablating the components (i.e., SBM and TNN), the ROC AUC score of SBM-TNN drops significantly. Our results indicate that community and topological information consistently boost the performance of link prediction.

**Computational Complexity** The topological complexity of the standard persistent homology (PH) matrix reduction algorithm runs in time at most $\mathcal{O}(Q^3)$, where $Q$ is the number of simplices in a filtration. For 0-dimensional PH, it can be computed efficiently using disjoint sets with complexity $\mathcal{O}(Q\alpha^{-1}(Q))$, where $\alpha^{-1}(\cdot)$ is the inverse Ackermann function. In our study, for graph representation learning, we only consider dimension 0 (connected components) and dimension 1 (cycles) due to the fact that we cannot observe enough higher-order (sub)structures in target datasets. If we consider high-dimensional topological features, the time complexity will grow large with the worst-case complexity $\mathcal{O}(m^d)$ for $d$-dimensional topological features (where $m$ denotes the number of edges).

## 7 Conclusion

We propose a new Stochastic Block Model-Aware Topological Neural Networks (SBM-TNN) method for both link prediction and knowledge graph completion tasks. By leveraging the topological information and estimated probability matrix with communities from different network topologies, SBM-TNN achieves state-of-the-art results on all datasets and the experimental evaluation confirms that SBM-TNN is accurate, flexible, and scalable. We also provide a theoretical guarantee for statistical inference based on the estimated

Table 5: Ablation studies.

| Architecture | Cora-ML | Citeseer | PubMed |
|---|---|---|---|
| SBM-TNN | **96.2±0.2** | **97.1±0.3** | **98.2±0.1** |
| SBM-GNN | 90.7±0.3 | 83.7±0.4 | 93.7±0.2 |
| TNN | 93.2±0.5 | 96.0±0.5 | 95.3±0.6 |

edge probabilities. Interesting future directions include extending how the SBM-TNN can be used for non-attributed, overlapping, dynamic network community detection, high-quality synthetic graph generation.

**Acknowledgments**

The research of Ma was supported in part by the U.S. National Science Foundation grants DMS-23-10288 and SES-24-12922. The research of Guo was supported in part by the National Natural Science Foundation of China grant (12301384, 12326615), and the Major Key Project of PCL (No. PCL2024A06).

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

## A   Derivation of Estimators under Undirected Network Models

Under the SBMs, to estimate $\boldsymbol{B}$, we note that for $1 \leq q < l \leq K$,

$$\boldsymbol{B}_{ql} := \frac{\sum_{1 \leq i \neq j \leq N} \boldsymbol{P}_{ij} \boldsymbol{Z}_{iq} \boldsymbol{Z}_{jl}}{\sum_{1 \leq i \neq j \leq N} \boldsymbol{Z}_{iq} \boldsymbol{Z}_{jl}} = \frac{\sum_{g_i=q,g_j=l} \boldsymbol{P}_{ij}}{N_q N_l} \quad \text{and} \quad \boldsymbol{B}_{qq} := \frac{\sum_{g_i=q,g_j=q} \boldsymbol{P}_{ij}}{N_q(N_q - 1)}.$$

Thus, it is reasonable to estimate $\boldsymbol{B}$ by the following $\hat{\boldsymbol{B}} = (\hat{\boldsymbol{B}}_{ql})_{1 \leq q \leq l \leq K}$,

$$\hat{\boldsymbol{B}}_{ql, q \neq l} := \frac{\sum_{\hat{g}_i = q, \hat{g}_j = l} \boldsymbol{A}_{ij}}{\hat{N}_q \hat{N}_l} \qquad \text{and} \qquad \hat{\boldsymbol{B}}_{qq} := \frac{\sum_{\hat{g}_i = q, \hat{g}_j = q} \boldsymbol{A}_{ij}}{\hat{N}_q (\hat{N}_q - 1)}.$$

Under the DCSBMs, to estimate $\boldsymbol{B}$, we note that

$$\sum_{g_i = q, g_j = l} \boldsymbol{P}_{ij} = \sum_{g_i = q} \boldsymbol{\theta}_i \sum_{g_j = l} \boldsymbol{\theta}_j \boldsymbol{B}_{ql} = N_q N_l \boldsymbol{B}_{ql},$$

Hence, we can estimate $\boldsymbol{B}$ by $\hat{\boldsymbol{B}} = (\hat{\boldsymbol{B}}_{ql})_{1 \leq q, l \leq K}$,

$$\hat{\boldsymbol{B}}_{ql} := \frac{\sum_{1 \leq i \neq j \leq N} \boldsymbol{A}_{ij} \hat{\boldsymbol{Z}}_{iq} \hat{\boldsymbol{Z}}_{jl}}{\sum_{1 \leq i \neq j \leq N} \hat{\boldsymbol{Z}}_{iq} \hat{\boldsymbol{Z}}_{jl}} = \frac{\sum_{\hat{g}_i = q, \hat{g}_j = l} \boldsymbol{A}_{ij}}{\hat{N}_q \hat{N}_l}.$$

To estimate $\boldsymbol{\theta}$, we note that

$$\sum_j \boldsymbol{P}_{ij} = \boldsymbol{\theta}_i \sum_k \sum_{g_j = k} \boldsymbol{\theta}_j \boldsymbol{B}_{g_i g_j} = \boldsymbol{\theta}_i \sum_k N_k \boldsymbol{B}_{g_i k} = \boldsymbol{\theta}_i \sum_{g_l = g_i} \sum_{j=1}^N \boldsymbol{P}_{lj} / N_{g_i},$$

where we used $\boldsymbol{B}_{g_i k} = \sum_{g_l = g_i, g_j = k} \boldsymbol{P}_{lj} / (N_{g_i} N_k)$. Hence, we estimate $\boldsymbol{\theta}_i$ by $\hat{\boldsymbol{\theta}}_i$ defined as

$$\hat{\boldsymbol{\theta}}_i = \frac{\hat{N}_{\hat{g}_i} \sum_j \boldsymbol{A}_{ij}}{\sum_{\hat{g}_l = \hat{g}_i} \sum_{j=1}^N \boldsymbol{A}_{lj}},$$

where $\hat{N}_{\hat{g}_i}$ is the number of nodes in the estimated community $\hat{g}_i$. We also denote $\hat{\boldsymbol{\Theta}} = \text{diag}(\hat{\boldsymbol{\theta}})$.

# B Proofs and Lemmas

For simplicity, we provide the proofs for the undirected network models SBMs and DCSBMs. In most cases, the proofs for the directed network models follow similarly. We highlight the differences if any.

## B.1 Proof of Lemma 5.6

*Proof.* First, we consider the strong consistency of community detection. Under the undirected network model SBM, the strong consistency of $\hat{\boldsymbol{Z}}$ follows from the Corollary II.1 in Su et al. (2019) and Assumptions 5.1 - 5.3. Under the undirected network model DCSBM, the strong consistency of $\hat{\boldsymbol{Z}}$ follows from Corollary III.1 in Su et al. (2019), where they considered the regularized DCSBMs with regularization parameter $\tau$. In our set-up, $\tau = 0$. Assumptions 5.1 - 5.4 imply that Assumptions 11-13 in Su et al. (2019) are satisfied. Hence the result follows. Under the directed network model ScBM, the strong consistency of $\hat{\boldsymbol{Y}}$ is similarly derived as that under the SBM. The strong consistency of $\hat{\boldsymbol{Z}}$ mainly depends on the success of Theorem II.1 in Su et al. (2019). In particular, Assumption 5.5 (with the notation simplified to the ScBMs) implies that the rows of the population singular vectors $\boldsymbol{V} = (v_i) \in \mathbb{R}^{n \times K^z}$ satisfies that

$$n^{1/2} \|v_i - v_j\|_2 \geq C > 0$$

for $g_i^z \neq g_j^z$ and some constant $C > 0$ (see details in Theorem 2.1 in the first version of Su et al. (2019)). Hence, Theorem II.1 in Su et al. (2019) holds and the strong consistency of $\hat{\boldsymbol{Z}}$ follows from Corollary II.1 in Su et al. (2019). Under the directed network model DCScBM, the strong consistency of $\hat{\boldsymbol{Y}}$ is similarly derived as that under the DCSBM. The strong consistency of $\hat{\boldsymbol{Z}}$ mainly depends on the success of Theorem III.4 in Su et al. (2019). In particular, Assumption 5.5 implies that $L_2$ normalized rows of the population singular vectors $\boldsymbol{V} = (v_i) \in \mathbb{R}^{n \times K^z}$ satisfies that

$$\left\| \frac{v_i}{\|v_i\|_2} - \frac{v_j}{\|v_j\|_2} \right\|_2 \geq C' > 0$$

for $g_i^z \neq g_j^z$ and some constant $C' > 0$ Rohe et al. (2016). Hence, Theorem III.4 in Su et al. (2019) holds and the strong consistency of $\hat{\boldsymbol{Z}}$ follows from Corollary III.1 in Su et al. (2019).

The strong consistency of $\hat{\boldsymbol{\theta}}$ (resp. $\hat{\boldsymbol{\theta}}^y$ and $\hat{\boldsymbol{\theta}}^z$) under DCSBMs (resp. DCScBMs) follows from Theorem III.6 in Su et al. (2019). Actually, the strong consistency of $\hat{\boldsymbol{Z}}$ (resp. $\hat{\boldsymbol{Y}}$ and $\hat{\boldsymbol{Z}}$) and Assumption 5.4 implies Assumption 15 in Su et al. (2019). And the result follows by noting that the minimal average degree in our set-up is $O(N\rho)$.

Now we proceed to show the strong consistency of $\hat{\boldsymbol{B}}$. The proof holds for both undirected and directed network models. For simplicity, we use the notation under undirected network models. For an $\epsilon_N$ to be selected, we have

$$\mathbb{P}\left(\sup_{1\leq q<l\leq K}|\hat{\boldsymbol{B}}_{ql} - \boldsymbol{B}_{ql}| \geq \epsilon_N \quad i.o.\right)$$

$$\leq \mathbb{P}\left(\sup_{1\leq q<l\leq K}|\hat{\boldsymbol{B}}_{ql} - \boldsymbol{B}_{ql}| \geq \epsilon_N \quad i.o., \sup_{1\leq i\leq N}\mathbf{1}\{\hat{g}_i \neq g_i\} = 0\right) + \mathbb{P}\left(\sup_{1\leq i\leq N}\mathbf{1}\{\hat{g}_i \neq g_i\} > 0 \quad i.o.\right)$$

$$\leq \mathbb{P}\left(\sup_{1\leq q<l\leq K}|\sum_{g_i=q,g_j=l}(\frac{\boldsymbol{A}_{ij}}{N_q N_l} - \frac{\boldsymbol{P}_{ij}}{N_q N_l})| \geq \epsilon_N \quad i.o.\right), \tag{13}$$

where the last inequality follows from the strong consistency of $\hat{g}_i$'s. To make the RHS of equation 13 zero, it suffices to show that

$$\sum_{K=1}^{\infty} \sum_{1\leq q<l\leq K} \mathbb{P}\left(|\sum_{g_i=q,g_j=l}(\frac{\boldsymbol{A}_{ij}}{N_q N_l} - \frac{\boldsymbol{P}_{ij}}{N_q N_l}))| \geq \epsilon_N\right) < \infty \tag{14}$$

for some $\epsilon_N$. To this end, we use the Bernstein inequality. Define $X^{(ij)} = \frac{\boldsymbol{A}_{ij}-\boldsymbol{P}_{ij}}{N_q N_l}$, we have $E(X^{(ij)}) = 0$, $|X^{(ij)}| \leq \frac{1}{N_q N_l}$ and

$$\sum_{g_i=q,g_j=l}\mathbb{E}[(X^{(ij)})^2] = \sum_{g_i=q,g_j=l}\mathbb{E}[\frac{(\boldsymbol{A}_{ij}-\boldsymbol{P}_{ij})^2}{N_q^2 N_l^2}] = \sum_{g_i=q,g_j=l}\frac{\boldsymbol{P}_{ij}(1-\boldsymbol{P}_{ij})}{N_q^2 N_l^2} = \frac{\boldsymbol{B}_{ql}(1-\boldsymbol{B}_{ql})}{N_q N_l}.$$

Then by the Bernstein equality, we have

$$\mathbb{P}(|\sum_{g_i=q,g_j=l}X^{(ij)}| \geq \epsilon_N) \leq \exp\left(-\frac{\frac{1}{2}\epsilon_N^2}{\frac{\boldsymbol{B}_{ql}(1-\boldsymbol{B}_{ql})}{N_q N_l} + \frac{\epsilon_N}{3N_q N_l}}\right).$$

Choosing $\epsilon_N = C\max_{ql}\sqrt{\frac{\boldsymbol{B}_{ql}(1-\boldsymbol{B}_{ql})}{N_q N_l}} \cdot \sqrt{\log N} = O(\frac{\sqrt{\rho\log N}}{N})$, it is easy to see that

$$\boldsymbol{B}_{ql}(1-\boldsymbol{B}_{ql}) \asymp \rho \gtrsim \frac{\sqrt{\rho\log N}}{N} \asymp \epsilon_N,$$

where the inequality follows from Assumption 5.2. We thus have

$$\mathbb{P}(|\sum_{g_i=q,g_j=l}X^{(ij)}| \geq \epsilon_N) \leq N^{-\alpha}$$

for some constant $\alpha > 0$. As a result, equation 14 is met because of fixed $K$. Finally, we obtain the strong consistency of $\hat{\boldsymbol{B}}$ that

$$\sup_{1\leq q<l\leq K}|\hat{\boldsymbol{B}}_{ql} - \boldsymbol{B}_{ql}| = O_{a.s.}(\frac{\sqrt{\rho\log N}}{N}).$$

The proof for $q = l$ goes similarly by noting that we can represent $\hat{\boldsymbol{B}}_{qq} - \boldsymbol{B}_{qq}$ as the following summation of independent terms,

$$\hat{\boldsymbol{B}}_{qq} - \boldsymbol{B}_{qq} = \sum_{g_i=q,g_j=q,i<j}\frac{\boldsymbol{A}_{ij}-\boldsymbol{P}_{ij}}{N_q(N_q-1)/2}.$$

Hence we omit it. □

## B.2 Proof of Theorem 5.8

*Proof.* We first provide the proof for the more general model DCSBMs, and then present the results for SBMs as a special case.

It is easy to see that

$$\hat{\boldsymbol{P}} - \boldsymbol{P} = \hat{\boldsymbol{\Theta}}\hat{\boldsymbol{Z}}\hat{\boldsymbol{B}}\hat{\boldsymbol{Z}}^\intercal\hat{\boldsymbol{\Theta}} - \boldsymbol{P} = (\hat{\boldsymbol{\Theta}} - \boldsymbol{\Theta})\hat{\boldsymbol{Z}}\hat{\boldsymbol{B}}\hat{\boldsymbol{Z}}^\intercal\hat{\boldsymbol{\Theta}} + \boldsymbol{\Theta}\hat{\boldsymbol{Z}}\hat{\boldsymbol{B}}\hat{\boldsymbol{Z}}^\intercal(\hat{\boldsymbol{\Theta}} - \boldsymbol{\Theta}) + \boldsymbol{\Theta}\hat{\boldsymbol{Z}}\hat{\boldsymbol{B}}\hat{\boldsymbol{Z}}^\intercal(\boldsymbol{\Theta}) - \boldsymbol{P}$$
$$:= I + II + III - \boldsymbol{P}. \tag{15}$$

Now we proceed to bound $I$. We have by Lemma 5.6 that

$$\sup_i |\hat{\boldsymbol{\theta}}_i - \boldsymbol{\theta}_i| = O_{a.s.}(\log N/(N\rho)),$$

and thus

$$\sup_i \hat{\boldsymbol{\theta}}_i \leq \sup_i |\hat{\boldsymbol{\theta}}_i - \boldsymbol{\theta}_i| + \sup_i \boldsymbol{\theta}_i = O_{a.s.}(1)$$

by noting Assumption 5.4. By Eqs. 8 and 9 in Lemma 5.6, We can easily have

$$\sup_{\hat{g}_i,\hat{g}_j}\hat{\boldsymbol{B}}_{\hat{g}_i\hat{g}_j} =_{a.s.} \sup_{g_i,g_j}\hat{\boldsymbol{B}}_{g_ig_j} \leq \sup_{g_i,g_j}|\hat{\boldsymbol{B}}_{g_ig_j} - \boldsymbol{B}_{g_ig_j}| + \sup_{g_i,g_j}\boldsymbol{B}_{g_ig_j}$$
$$= O_{a.s.}(\frac{\sqrt{\rho\log N}}{N}) + O_{a.s.}(\rho) = O_{a.s.}(\rho),$$

where the last equality is implied by Assumption 5.2. As a result,

$$\sup_{1\leq i,j\leq N}|(I)_{ij}| = \sup_{1\leq i,j\leq N}|(\mathrm{diag}(\hat{\boldsymbol{\theta}} - \boldsymbol{\theta})\hat{\boldsymbol{Z}}\hat{\boldsymbol{B}}\hat{\boldsymbol{Z}}^\intercal\mathrm{diag}(\hat{\boldsymbol{\theta}}))_{ij}|$$
$$= \sup_{1\leq i,j\leq N}|(\hat{\boldsymbol{\theta}}_i - \boldsymbol{\theta}_i)\hat{\boldsymbol{B}}_{\hat{g}_i\hat{g}_j}\hat{\boldsymbol{\theta}}_j| = O_{a.s.}(\log N/N). \tag{16}$$

Similarly, we can show that

$$\sup_{1\leq i,j\leq N}|(II)_{ij}| = O_{a.s.}(\log N/N). \tag{17}$$

It remains to bound $III - \boldsymbol{P}$. Noting

$$\boldsymbol{\Theta}\hat{\boldsymbol{Z}}\hat{\boldsymbol{B}}\hat{\boldsymbol{Z}}^\intercal\boldsymbol{\Theta} - \boldsymbol{P} = \boldsymbol{\Theta}(\hat{\boldsymbol{Z}}\hat{\boldsymbol{B}}\hat{\boldsymbol{Z}}^\intercal - \boldsymbol{Z}\boldsymbol{B}\boldsymbol{Z}^\intercal)\boldsymbol{\Theta} \tag{18}$$

and the boundness of $\boldsymbol{\theta}$ by Assumption 5.4, we only need to bound $\hat{\boldsymbol{Z}}\hat{\boldsymbol{B}}\hat{\boldsymbol{Z}}^\intercal - \boldsymbol{Z}\boldsymbol{B}\boldsymbol{Z}^\intercal$. It is easy to observe that

$$\hat{\boldsymbol{Z}}\hat{\boldsymbol{B}}\hat{\boldsymbol{Z}}^\intercal - \boldsymbol{Z}\boldsymbol{B}\boldsymbol{Z}^\intercal = (\hat{\boldsymbol{Z}} - \boldsymbol{Z})\boldsymbol{B}\hat{\boldsymbol{Z}}^\intercal + \boldsymbol{Z}\boldsymbol{B}(\hat{\boldsymbol{Z}} - \boldsymbol{Z})^\intercal + \hat{\boldsymbol{Z}}(\hat{\boldsymbol{B}} - \boldsymbol{B})\hat{\boldsymbol{Z}}^\intercal$$
$$:= E_1 + E_2 + E_3.$$

For $E_3$, we further have

$$E_3 = \hat{\boldsymbol{Z}}(\hat{\boldsymbol{B}} - \boldsymbol{B})\hat{\boldsymbol{Z}}^\intercal$$
$$= \boldsymbol{Z}(\hat{\boldsymbol{B}} - \boldsymbol{B})\boldsymbol{Z}^\intercal + (\hat{\boldsymbol{Z}} - \boldsymbol{Z})(\hat{\boldsymbol{B}} - \boldsymbol{B})\hat{\boldsymbol{Z}}^\intercal + \boldsymbol{Z}(\hat{\boldsymbol{B}} - \boldsymbol{B})(\hat{\boldsymbol{Z}} - \boldsymbol{Z})^\intercal.$$

Combining the above two facts, we obtain

$$\hat{\boldsymbol{Z}}\hat{\boldsymbol{B}}\hat{\boldsymbol{Z}}^\intercal - \boldsymbol{Z}\boldsymbol{B}\boldsymbol{Z}^\intercal = \boldsymbol{Z}(\hat{\boldsymbol{B}} - \boldsymbol{B})\boldsymbol{Z}^\intercal + R, \tag{19}$$

where

$$R := (\hat{\boldsymbol{Z}} - \boldsymbol{Z})\boldsymbol{B}\hat{\boldsymbol{Z}}^\intercal + \boldsymbol{Z}\boldsymbol{B}(\hat{\boldsymbol{Z}} - \boldsymbol{Z})^\intercal + (\hat{\boldsymbol{Z}} - \boldsymbol{Z})(\hat{\boldsymbol{B}} - \boldsymbol{B})\hat{\boldsymbol{Z}}^\intercal + \boldsymbol{Z}(\hat{\boldsymbol{B}} - \boldsymbol{B})(\hat{\boldsymbol{Z}} - \boldsymbol{Z})^\intercal.$$

By the strong consistency of $\hat{\boldsymbol{Z}}$,

$$\hat{\boldsymbol{Z}} - \boldsymbol{Z} = \boldsymbol{0}, \quad a.s., \tag{20}$$

and thus $R =_{a.s.} \boldsymbol{0}$. It remains to bound $\boldsymbol{Z}(\hat{\boldsymbol{B}} - \boldsymbol{B})\boldsymbol{Z}^{\intercal}$. Noting

$$(\boldsymbol{Z}(\hat{\boldsymbol{B}} - \boldsymbol{B})\boldsymbol{Z}^{\intercal})_{ij} = (\hat{\boldsymbol{B}} - \boldsymbol{B})_{g_i g_j}$$

and equation 8 in Lemma 5.6, we have

$$\sup_{i,j} |\boldsymbol{Z}(\hat{\boldsymbol{B}} - \boldsymbol{B})\boldsymbol{Z}^{\intercal})|_{ij} = O_{a.s.}(\frac{\sqrt{\rho \log N}}{N}).$$

As a result, we have

$$\sup_{1 \le i,j \le N} |(III - \boldsymbol{P})_{ij}| = O_{a.s.}(\frac{\sqrt{\rho \log N}}{N}). \tag{21}$$

Combining equation 16, equation 17 with equation 21, we have for DCSBMs that

$$\sup_{1 \le i,j \le N} |\hat{\boldsymbol{P}}_{ij} - \boldsymbol{P}_{ij}| = O_{a.s.}(\frac{\log N}{N}). \tag{22}$$

While for SBMs, the error terms $I$ and $II$ are exactly zero. Hence,

$$\sup_{1 \le i,j \le N} |\hat{\boldsymbol{P}}_{ij} - \boldsymbol{P}_{ij}| = O_{a.s.}(\frac{\sqrt{\rho \log N}}{N}).$$

The proofs for the directed network models ScBMs and DCScBMs go similarly provided that Assumption 5.5 is additionally required for Lemma 5.6. $\square$

## B.3 Proof of Theorem 5.8

*Proof.* Recall equation 15, we can decompose $\hat{\boldsymbol{P}} - \boldsymbol{P}$ by $I + II + III - \boldsymbol{P}$. For DCSBMs, we have shown in the proof of Theorem 5.7 that $\sup|(I)_{ij}| = O_{a.s.}(\log N/N)$ and $\sup|(II)_{ij}| = O_{a.s.}(\log N/N)$. For SBMs, $I = II = \boldsymbol{0}$.

We now show the asymptotic normality of $III - \boldsymbol{P}$, for which by equation 18, equation 19 and equation 20, we only need to show the asymptotic normality of $\boldsymbol{Z}(\hat{\boldsymbol{B}} - \boldsymbol{B})\boldsymbol{Z}^{\intercal}$. Note that

$$(\boldsymbol{Z}(\hat{\boldsymbol{B}} - \boldsymbol{B})\boldsymbol{Z}^{\intercal})_{ij} = (\hat{\boldsymbol{B}} - \boldsymbol{B})_{g_i g_j} = \hat{\boldsymbol{B}}_{ql} - \boldsymbol{B}_{ql}$$

provided that $g_i = q$ and $g_j = l$. By strong consistency of $\hat{\boldsymbol{Z}}$, we have $\hat{g}_i =_{a.s.} g_i$ and $\hat{N}_q =_{a.s.} N_q$ for all $i \in [N]$ and $q \in [K]$. So for $q \ne l$,

$$\hat{\boldsymbol{B}}_{ql} - \boldsymbol{B}_{ql} = \sum_{g_i = q, g_j = l} \frac{\boldsymbol{A}_{ij} - \boldsymbol{P}_{ij}}{N_q N_l}.$$

We use he Lindeberg-Feller Central Limit Theorem to derive the limit distribution of $\hat{\boldsymbol{B}}_{ql} - \boldsymbol{B}_{ql}$. First, note that

$$s_N^2 := \mathrm{Var}(\sum_{g_i = q, g_j = l} \frac{\boldsymbol{A}_{ij} - \boldsymbol{P}_{ij}}{N_q N_l}) = \sum_{g_i = q, g_j = l} \mathrm{Var}(\frac{\boldsymbol{A}_{ij}}{N_q N_l})$$

$$= \sum_{g_i = q, g_j = l} \boldsymbol{P}_{ij}(1 - \boldsymbol{P}_{ij})/(N_q^2 N_l^2) = \boldsymbol{B}_{ql}(1 - \boldsymbol{B}_{ql})/(N_q N_l)$$

We only need to show

$$\frac{1}{s_N^2} \sum_{g_i = q, g_j = l} \mathbb{E}\{(\frac{\boldsymbol{A}_{ij} - \boldsymbol{P}_{ij}}{N_q N_l}))^2 \mathbb{I}(|\frac{\boldsymbol{A}_{ij} - \boldsymbol{P}_{ij}}{N_q N_l})| \ge \epsilon s_N)\} \to 0$$

for every $\epsilon > 0$, which holds sufficiently if $|\frac{\boldsymbol{A}_{ij} - \boldsymbol{P}_{ij}}{N_q N_l})| \leq \epsilon s_N$. Indeed,

$$|\frac{\boldsymbol{A}_{ij} - \boldsymbol{P}_{ij}}{N_q N_l})| \lesssim \frac{1}{N^2} \lesssim \frac{\rho^{1/2}}{N} \asymp \epsilon\sqrt{\boldsymbol{B}_{ql}(1 - \boldsymbol{B}_{ql})/N_q N_l}.$$

Therefore, by central limit theorem, we can obtain in SBMs that

$$N(\hat{\boldsymbol{B}}_{ql} - \boldsymbol{B}_{ql}) \to N(0, \frac{\boldsymbol{B}_{ql}(1 - \boldsymbol{B}_{ql})}{\pi_q \pi_l})$$

for $q \neq l$. When $q = l$, considering the dependency of $\boldsymbol{A}_{ij}$ and $\boldsymbol{A}_{ji}$, we consider the following halved term

$$\hat{\boldsymbol{B}}_{qq} - \boldsymbol{B}_{qq} = \sum_{g_i = q, g_j = q, i < j} \frac{\boldsymbol{A}_{ij} - \boldsymbol{P}_{ij}}{N_q(N_q - 1)/2}. \tag{23}$$

Similarly, we can derive

$$N(\hat{\boldsymbol{B}}_{qq} - \boldsymbol{B}_{qq}) \to N(0, \frac{2\boldsymbol{B}_{qq}(1 - \boldsymbol{B}_{qq})}{\pi_q^2}).$$

For DCSBMs, we thus have that

$$N(\hat{\boldsymbol{P}} - \boldsymbol{P} + \boldsymbol{E})_{ij} \to N(0, \boldsymbol{\theta}_i^2 \cdot \frac{\boldsymbol{B}_{g_i g_j}(1 - \boldsymbol{B}_{g_i g_j})}{\pi_{g_i} \pi_{g_j}} \cdot \boldsymbol{\theta}_j^2)$$

with $\boldsymbol{E}_{ij} = O_{a.s.}(\log N/N)$ and $g_i \neq g_j$, and

$$N(\hat{\boldsymbol{P}} - \boldsymbol{P} + \boldsymbol{F})_{ij} \to N(0, 2\boldsymbol{\theta}_i^2 \cdot \frac{\boldsymbol{B}_{g_i g_i}(1 - \boldsymbol{B}_{g_i g_i})}{\pi_{g_i}^2} \cdot \boldsymbol{\theta}_j^2)$$

with $\boldsymbol{F}_{ij} = O_{a.s.}(\log N/N)$ and $g_i = g_j$.

The proofs for the directed network models ScBMs and DCScBMs are similar except that when $g_i^y = g_j^z$, the directed networks do not involve dependency of pairs $\boldsymbol{A}_{ij}$ and $\boldsymbol{A}_{ji}$ like in (23), hence the constant 2 in the asymptotic variances are replaced by 1.

$$\square$$

## B.4 Auxiliary lemmas

**Lemma B.1** (Bernstein inequality). *Let $X_1, \ldots, X_N$ be independent zero-mean random variables. Suppose that $|X_i| \leq M$ almost surely, for all $i$. Then, for all positive $t$,*

$$\mathbb{P}\left(\sum_{i=1}^N X_i \geq t\right) \leq \exp\left(-\frac{\frac{1}{2}t^2}{\sum_{i=1}^N \mathbb{E}[X_i^2] + \frac{1}{3}Mt}\right).$$

## C Notes, Notations, and Flowchart

The notations are summarized in Table 6.

## D Additional Experiment Settings

For all 5 datasets (i.e., Cora-ML, Citeseer, PubMed, Photo, and Computers), SBM-TNN is trained by the Adam optimizer with the Cross Entropy Loss function. Additionally, all baseline methods are initialized with the parameters suggested in their respective works, we carefully tune the parameters during training to ensure that the baseline model achieves optimal performance. Here we treat the resulting topological features in dimension 0 (connected components) and 1 (cycles) (i.e., $Q = 2$). For link prediction, we perform an extensive

Table 6: A summary of the notes and notations.

| Notation | Definition |
|---|---|
| **Notations for undirected network models:** | |
| $N$ | Number of communities |
| $g_i \in \{1, ..., K\}$ | Community assignment of node $i$ |
| $N_k$ | Number of nodes within the $k$-th community |
| $\pi_k := N_k/N$ | Proportion of nodes in the $k$-th community |
| $\boldsymbol{Z} \in \{0,1\}^{N \times K}$ | Membership matrix |
| $\boldsymbol{B} \in \mathbb{R}^{K \times K}$ | Block probability matrix |
| $\boldsymbol{\theta} \in \mathbb{R}^N$ | Node propensity vector |
| $\boldsymbol{\Theta} = \text{diag}\{\boldsymbol{\theta}_1, ..., \boldsymbol{\theta}_N\}$ | Node propensity matrix |
| $\boldsymbol{P} := \boldsymbol{\Theta Z B Z^\intercal \Theta} \in \mathbb{R}^{N \times N}$ | Edge probability matrix |
| $d_i = \sum_{j=1}^N \boldsymbol{P}_{ij}$ | Population degree of node $i$ |
| $\mathcal{D} = \text{diag}\{d_1, ..., d_N\}$ | Population degree matrix |
| $\rho$ | Network sparsity |
| $\boldsymbol{A} \in \{0,1\}^{N \times N}$ | Symmetric Adjacency matrix |
| $\hat{d}_i = \sum_{j=1}^N \boldsymbol{A}_{ij}$ | Degree of node $i$ |
| $\boldsymbol{D} = \text{diag}\{\hat{d}_1, ..., \hat{d}_N\}$ | Degree matrix |
| $\boldsymbol{L} = \boldsymbol{I} + \boldsymbol{D}^{-1/2} \boldsymbol{A} \boldsymbol{D}^{-1/2}$ | Graph Laplacian |
| **Notations for directed network models:** | |
| $N$ | Number of nodes |
| $K^y (K^z)$ | Number of row (column) clusters |
| $g_i^y \in \{1, ..., K^y\}(g_i^z \in \{1, ..., K^z\})$ | Row (column) cluster assignment of node $i$ |
| $N_k^y(N_k^z)$ | Number of nodes in the $k^y$th ($k^z$th) row (column) cluster |
| $\pi_k^y := N_k^y/N(\pi_k^z := N_k^z/N)$ | Proportion of nodes in the $k^y$th ($k^z$th) row (column) cluster |
| $\boldsymbol{Y} \in \{0,1\}^{N \times K^y}(\boldsymbol{Z} \in \{0,1\}^{N \times K^z})$ | Row (Column) membership matrix |
| $\boldsymbol{B} \in \mathbb{R}^{K^y \times K^z}(K^y \leq K^z)$ | Block probability matrix |
| $\boldsymbol{\theta}^y \in \mathbb{R}^N(\boldsymbol{\theta}^z \in \mathbb{R}^N)$ | Node propensity vector for sending (receiving) edges |
| $\boldsymbol{\Theta}^y = \text{diag}\{\boldsymbol{\theta}_i^y\}(\boldsymbol{\Theta}^z = \text{diag}\{\boldsymbol{\theta}_i^z\})$ | Node propensity matrix for sending (receiving) edges |
| $\boldsymbol{P} := \boldsymbol{\Theta}^y \boldsymbol{Y} \boldsymbol{B} \boldsymbol{Z}^\intercal \boldsymbol{\Theta}^z \in \mathbb{R}^{N \times N}$ | Edge probability matrix |
| $d_i^y = \sum_{j=1}^N \boldsymbol{P}_{ij}(d_i^z = \sum_{j=1}^N \boldsymbol{P}_{ji})$ | Population out-degree (in-degree) of node $i$ |
| $\mathcal{D}^y = \text{diag}\{d_1^y, ..., d_N^y\}(\mathcal{D}^z = \text{diag}\{d_1^z, ..., d_N^z\})$ | Population out-degree (in-degree) matrix |
| $\rho$ | Network sparsity |
| $\boldsymbol{A} \in \{0,1\}^{N \times N}$ | Asymmetric adjacency matrix |
| $\hat{d}_i^y = \sum_{j=1}^N \boldsymbol{A}_{ij}(\hat{d}_i^z = \sum_{j=1}^N \boldsymbol{A}_{ji})$ | Out-degree (in-degree) of node $i$ |
| $\boldsymbol{D}^y = \text{diag}\{\hat{d}_1^y, ..., \hat{d}_N^y\}(\boldsymbol{D}^z = \text{diag}\{\hat{d}_1^z, ..., \hat{d}_N^z\})$ | Out-degree (In-degree) matrix |
| $\boldsymbol{L} = \boldsymbol{I} + (\boldsymbol{D}^y)^{-1/2} \boldsymbol{A} (\boldsymbol{D}^z)^{-1/2}$ | Graph Laplacian |

grid search for learning rate among $\{0.001, 0.005, 0.008, 0.01, 0.1\}$, the dropout rate among $\{0.1, 0.2, \ldots, 0.9\}$, the number of hidden units among $\in \{8, 16, 32, 64, 128\}$, and the model is trained for 5,000 epochs with early stopping applied when the metric (i.e., validation loss) starts to drop. For KG completion, we set the batch size to be 512 and the model is trained for 500 epochs, and we perform an extensive grid search for learning rate among $\{0.00001, 0.001, 0.01, 0.1\}$.

# E   Additional Experiments

To better illustrate the benefits of involving community-level and topological information for link prediction tasks, we incorporate the SBM-TNN framework into ROLAND architecture You et al. (2022) for dynamic link prediction (in mean reciprocal rank (MRR)) on Bitcoin-OTC data. As shown in Table 7, we found that ROLAND + SBM-TNN achieves 17.80% performance gain over the ROLAND.

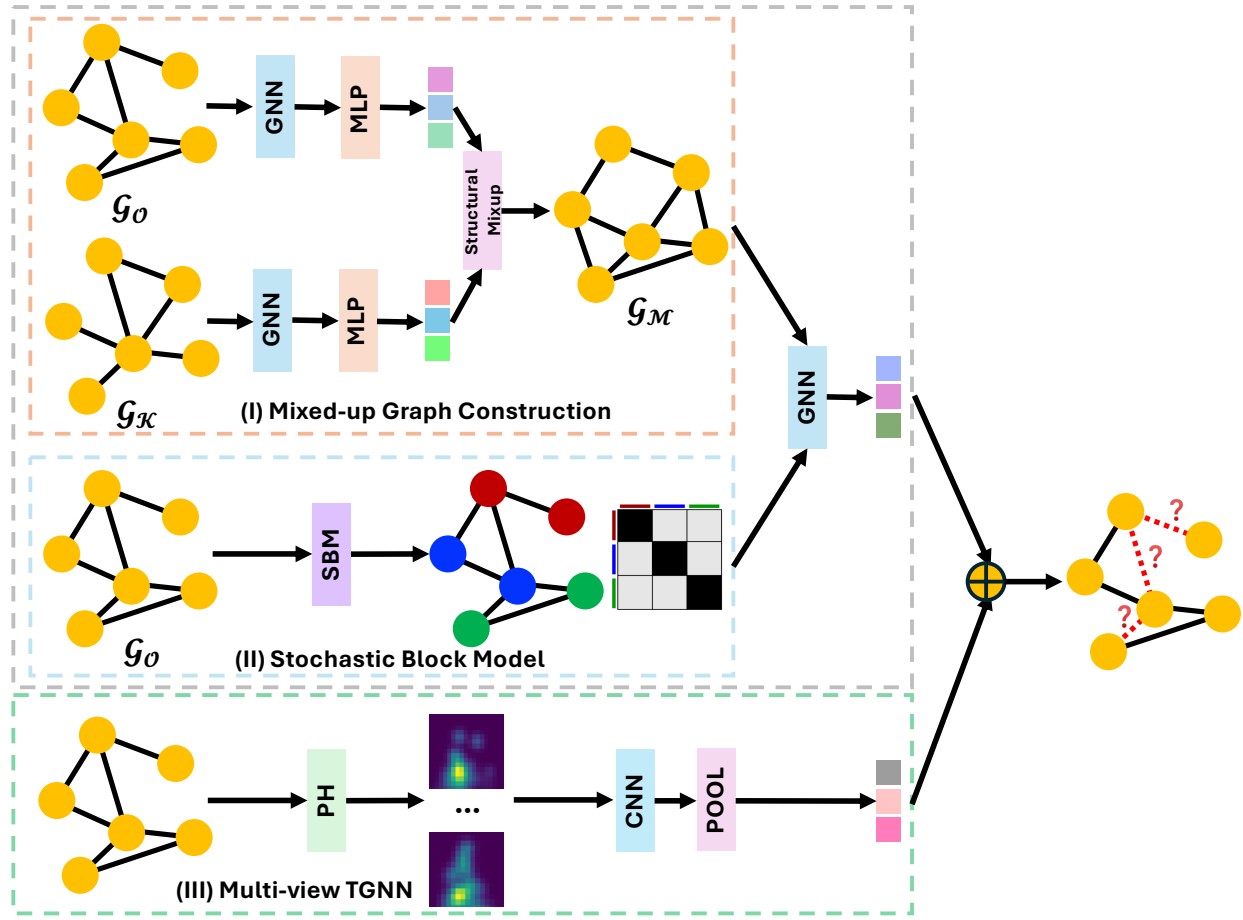

Figure 1: The overall architecture of SBM-TNN.

Table 7: Performance comparison on Bitcoin-OTC.

| Method | Mean Reciprocal Rank (MRR) |
|---|---|
| ROLAND | $0.2203 \pm 0.0167$ |
| **ROLAND + SBM-TNN (ours)** | **$0.2595 \pm 0.0246$** |

We have also conducted additional experiments for link prediction by comparing with the Edge-featured Graph Attention Network (EGAT) model Wang et al. (2021b). From the below table, we observe that SBM-TNN always achieves the highest AUC-ROC score with average relative gain of 3.78%.

Table 8: Performance comparison on link prediction.

| Method | Cora-ML | Citeseer | PubMed |
|---|---|---|---|
| EGAT | $92.3 \pm 0.6$ | $93.8 \pm 1.0$ | $94.8 \pm 0.3$ |
| **SBM-TNN (ours)** | **$96.2 \pm 0.2$** | **$97.1 \pm 0.3$** | **$98.2 \pm 0.1$** |

