# OpenReview forum: "Stochastic Block Model-Aware Topological Neural Networks for Graph Link Prediction"
_TMLR — Accepted by TMLR_

### Review · Reviewer_nftF · 2025-04-25

**Summary Of Contributions:**

This paper proposes SBM-TNN, a novel graph neural network framework that integrates Stochastic Block Models (SBMs) and topological descriptors (via persistent homology) to improve link prediction tasks on graphs and knowledge graphs. The key contributions include:

1) Incorporation of SBM-estimated edge probabilities in GNNs to better capture community structure.

2) Use of topological signatures (e.g., persistence images) to encode higher-order information.

3) Theoretical guarantees: Provides entrywise error bounds and asymptotic normality for the edge probability matrix.

4) Demonstrates state-of-the-art results on both graph and knowledge graph link prediction tasks.

**Audience:**

Yes

**Broader Impact Concerns:**

None concerns

**Claims And Evidence:**

Yes

**Requested Changes:**

- Visualization: Add a high-level diagram summarizing the architecture: SBM estimation, topological embedding, mixup graph, GNN  link prediction.

- Baselines: Include or mention models that use probabilistic edges or uncertainty, like variational GNNs or Bayesian GNNs.

- Efficiency Metrics: Report time/memory vs baselines to show practical benefits of using SBM-based priors.

- Model Generalization: Discuss whether this framework could extend to dynamic graphs, heterogeneous graphs, or non-relational data.

**Strengths And Weaknesses:**

Strengths:

1. Novel Integration of Concepts
The integration of SBM-based community detection with topological neural networks is novel and addresses key limitations of GNNs, such as difficulty in capturing higher-order structure and uncertainty.

2. Strong Theoretical Backbone
The paper provides rigorous statistical analysis of the edge probability estimator, including:

Entrywise bounds on the estimated probability matrix.

Asymptotic normality, enabling uncertainty quantification.

These are rare in deep learning-oriented graph papers and strengthen the theoretical contribution.

3. Empirical Performance
Shows consistent and significant improvements over strong baselines across citation networks (Cora-ML, Citeseer, PubMed), Amazon graphs, and KGs (FB15k-237, WN18RR, NELL-995).

Ablation studies validate the individual contributions of SBM and topology modules.

4. Multi-view Topological Learning
Uses multiple filtration functions (centrality scores) to build topological summaries, offering a richer representation space.

5. Solid Benchmarks
The model is compared against a wide array of modern GNN and KGC baselines (e.g., SEAL, TLC-GNN, NBFNet).


Weaknesses:

1. Clarity & Overcomplexity
The method section is dense, with multiple architectures (SBM estimator, mixed-up graph, persistence images, GNN layers). The pipeline could benefit from a clear diagram or step-by-step illustration.

Some notations are introduced with minimal explanation (e.g., Z, B̂, Θ̂), and repeated definitions for similar models (SBM, DCSBM, ScBM, DCScBM) could be streamlined.

2. Lack of Baselines Using Edge Probabilities
No direct comparison is made with other models that use probabilistic or edge-weighted input graphs, which would contextualize the benefit of using SBM-derived probabilities.

3. Efficiency / Scalability Discussion
While some complexity is discussed (e.g., persistent homology at 0- and 1-dimensions), there’s little exploration of scalability on larger graphs, e.g., OGB datasets (aside from a brief mention of ogbl-wikikg2). No training time, GPU usage, or memory benchmarks are provided.

4. Assumption Dependence in Theory
The strong theoretical guarantees (entrywise bounds, normality) rely on strong assumptions (e.g., balanced clusters, fixed K, bounded degree heterogeneity), which may limit real-world applicability.

---

> ### Author Response · Authors · 2025-05-14
> **Rebuttal by Authors**
>
> Dear Reviewer nftF, we sincerely thank you for your valuable feedback on our submission. Below is our responses to the concerns you mentioned. We have incorporated these contents into the updated version of our paper.
>
> **Q1:** A clear diagram.
>
> **A:** We have included the flowchart of the SBM-TNN model in Appendix C.
>
>
> **Q2:** Some notations are introduced with minimal explanation.
>
> **A:** Thank you for your helpful comments. We have added detailed explanations for the parameters in the context of SBMs, while the parameter descriptions for other models (DCSBMs, ScBM, DCScBM) have been streamlined.
>
> **Q3:** Lack of Baselines Using Edge Probabilities.
>
> **A:** Thank you very much for this question. We have conducted additional experiments for link prediction by comparing with the EGAT model (which proposes edge-featured graph attention layers) [1]. From the below table, we observe that SBM-TNN always achieves the highest AUC-ROC score with average relative gain of 3.78\%.
>
>
> ||Cora-ML|Citeseer|PubMed|
> |-|-|-|-|
> |EGAT|92.3±0.6 | 93.8±1.0| 94.8±0.3|
> |SBM-TNN| **96.2±0.2** | **97.1±0.3**| **98.2±0.1**|
>
> [1] Wang, Z., Chen, J. and Chen, H., 2021. EGAT: Edge-featured graph attention network. In Artificial Neural Networks and Machine Learning–ICANN International Conference on Artificial Neural Networks.
>
> **Q4:** No training time, GPU usage.
>
> **A:** All experiments in this paper were conducted on a Linux server with 2 NVIDIA A5000 GPUs. The average training time per epoch (in seconds) on Cora-ML, Citeseer, PubMed, FB15k-237, and ogbl-wikikg2 data are:
> ||Cora-ML|Citeseer|PubMed|FB15k-237| ogbl-wikikg2|
> |-|-|-|-|-|-|
> |SBM-TNN| 1.60 s | 1.43 s | 15.22 s | 25.73 s | 221.18 s|
>
> **Q5:** Assumption Dependence.
>
> **A:** Thank you for your concern and helpful comments. For “balanced clusters,” the original manuscript might be confusing. We actually allow the number of nodes in each cluster to differ, as long as they vary in the same order. This assumption is primarily imposed to guarantee the strong consistency of SBMs, which is crucial for establishing asymptotic normality, as it ensures that every node is correctly clustered. Without this assumption, nodes belonging to small clusters may not be accurately clustered.
>
> For fixed K, this assumption is made for notational simplicity. This can certainly be extended to accommodate varying numbers of communities, as in Su et al., 2019.
>
> For bounded degree heterogeneity, this assumption on the upper bound is mild, as the node propensity parameters are required to be normalized (see Equation (5) in the revision) to ensure model identifiability. For the lower bound, it can be relaxed to $n^{-\alpha}$ for some positive constant $\alpha$ with a sacrifice of the simplicity of other conditions.
>
> In the revision, we have revised Assumption 5.1  and have added remarks after the assumptions to explain the three assumptions.

---

### Review · Reviewer_rmD1 · 2025-04-27

**Summary Of Contributions:**

This paper proposes a stochastic block model-aware topologcal neural networks, named SBM-TNN. The proposed method is designed to learn the probabilities between pairs of nodes, which is acheved by replacing the discrete edges between nodes with probabilities. However, this idea has been widely used in the generative graph neural networks where the node embedding, clustering and edge properity are naturally correlated.

**Audience:**

Yes

**Broader Impact Concerns:**

No ethical implications.

**Claims And Evidence:**

No

**Requested Changes:**

1. The motivation of this paper should be corrected, the authors should discuss the difference between generative GNNs and the proposed method.
2. The technical contribution should be highlighted and clearly discussed, the current version is not very encouraging to me.
3. The experiments should be more comprehensive.
4. The paper written should be improved. The equations are indexed randomly, which should be carefully managed.

**Strengths And Weaknesses:**

Strength:
1. The paper is overall easy to follow.
2. The authors provide theoretical connections between the proposed ScBMs/DCScBMs and existing SBMs/DCSBMs.

Weakness:
1. The motivation of this paper is based on an over-claimed justification. The authors claim that existing GNNs 'can not efficiently learn the edge probabilities based on topological structures (i.e., higher-order interactions) and node features', which is absolutely over-claimed. There are many works that have the same motivation. Specifically, most of the generative GNNs are based on this motivation: the embeddings are related to topological structures and features, the edges probability is modeled by Bernoulli distribution related to the node embeddings. I suggest the authors catch up with the generative GNNs for better motivation.
2. The authos propose two variants of existing methods SBMs/DCSBMs, namely ScBMs/DCScBMs.However, the model structure is highly similar to exitsing works while the major difference lies in the multi-view part. However, the multi-view part is manually designed and I did not understand why it is necessary and effective. Currently, this combination seems like the ensemble learning. As a result, the technical contribution should be further highlighted.
3. The experiments are not extensive. Only a few tiny graphs are evaluated while the inference of this model is time consuming and the authors should provide both theoretical and experimental evaluation. Furthermore, as I mentioned, the baselines should include many generative GNNs to verify the effectiveness of the proposed method.

---

> ### Author Response · Authors · 2025-05-14
> **Rebuttal by Authors**
>
> Dear Reviewer rmD1, thanks very much for your valuable feedback on our submission. Below is our responses to the concerns and questions you raised. We have incorporated these contents into the updated version of our paper.
>
> **Q1:** The motivation of this paper.
>
> **A:** Thank you for this question. We would like to clarify that our work is focused on the link prediction tasks (including regular graphs and knowledge graphs), rather than on graph generation via generative GNN frameworks. Our contribution is not to introduce a new generative paradigm, but rather to improve representation learning for link prediction by explicitly modeling community-level information and local topological information using SBM and topological representation module, respectively. For link prediction, real-world networks are sparse with a large number of potential links but only a few actual connections, which makes it difficult to accurately predict missing links. Also, our target networks are large and complex with high-dimensional feature spaces, and they consist of different types of nodes with different characteristics. In this case, SBM-based methods can explicitly capture community structures within the network, i.e., nodes are assigned to communities, and connections are more likely within communities than between them. This helps in understanding the underlying structure and predicting links based on community membership. The proposed topological representation learning module goes beyond pairwise connections to analyze higher-order structures such as cycles and cliques, and these features can provide additional information for link prediction.
>
> **Q2:** Technical contribution should be further highlighted.
>
> **A:** Thank you for your comment and suggestion.
>
>  The main contributions of our paper are as follows:
> We propose SBM-TNN, the first model to integrate statistical modeling via Stochastic Block Models (SBMs) with topological learning for graph representation. To enhance the expressiveness of node embeddings, our model incorporates persistent homology and multi-view convolutional layers to extract high-order topological features—an aspect not captured by conventional GCNs. Additionally, we establish theoretical guarantees by deriving entrywise error bounds and proving the asymptotic normality of the estimated edge probability matrix. These results enable the construction of confidence intervals and provide a principled framework for uncertainty quantification, a feature largely absent in existing GNN models.
>
> Compared with [1], we use spectral methods instead of MCMC, and we have the asymptotic distribution of link prediction. Compared with [2], we use spectral methods, however, [2] uses Bayesian methods and only focus on node classification. Compared with [3], it focuses on node classification, however, our paper focuses on link prediction and knowledge graph completion. For our proposed SBM-TNN model, we use topological neural networks (TNN) instead of GNNs which allows us to incorporate both local and global topological information of a graph using persistent homology. Equipped with SBM, our model can encode both topological features and community information. Through the multi-view graph representation learning, the model can capture different views and highlight different structural or semantic aspects of the graph. In addition, the model becomes more robust to noise and the multi-view part helps avoid overfitting to one specific graph construction or neighborhood definition.
>
> [1] Mehta N, Duke L C, Rai P. Stochastic blockmodels meet graph neural networks. International Conference on Machine Learning. PMLR, 2019: 4466-4474.
>
> [2] Liu X, Yang B, Song W, et al. A block-based generative model for attributed network embedding. World Wide Web, 2021, 24: 1439-1464.
>
> [3] He D, Liang C, Liu H, et al. Block modeling-guided graph convolutional neural networks. Proceedings of the AAAI conference on artificial intelligence. 2022, 36(4): 4022-4029.
>
> **Q3:** The experiments are not extensive.
>
> **A:** In this paper, we conduct experiments on 9 different datasets for link prediction and knowledge graph completion tasks. To better illustrate the benefits of involving community-level and topological information for link prediction tasks, we incorporate the SBM-TNN framework into ROLAND [4] for dynamic link prediction (in mean reciprocal rank (MRR)) on Bitcoin-OTC data. As shown in the below table, we found that ROLAND + SBM-TNN achieves 17.80\% performance gain over the ROLAND.
>
> ||Bitcoin-OTC|
> |-|-|
> |ROLAND| 0.2203±0.0167|
> |ROLAND + SBM-TNN| **0.2595±0.0246**|
>
> We did not include generative GNNs, which are typically designed for graph generation. However, we have added a discussion in the conclusion to highlight potential connections between our proposed model and generative GNN approaches.
>
> [4] You, J., Du, T. and Leskovec, J., 2022, August. ROLAND: graph learning framework for dynamic graphs. ACM SIGKDD.

---

> > ### Comment · Reviewer_rmD1 · 2025-06-04
> > **Response**
> >
> > Thanks for the authors' response. However, I think my concerns are not addressed:
> >
> > For Q1 and Q2: I am not arguing the authors do not provide a clear motivation, my focus are on the over-claim: 'existing GNNs 'can not efficiently learn the edge probabilities based on topological structures (i.e., higher-order interactions) and node features', which is significantly not correct, I expect the authors directly explain on this rather than repeat the motivation.
> >
> > Q2: The necessary of multi-view is not well supported, either by experimental or theoretical results.
> >
> > Q3: Generative GNNs are not for graph generation, instead, they are designed for link prediction. Please carefully read the paper such as Variational Graph AutoEncoder(VGAE) and many following works.

---

### Review · Reviewer_LToV · 2025-04-29

**Summary Of Contributions:**

The authors propose a novel stochastic block model (SBM)-aware topological neural networks, called SBM-TNN. It uses SBMs to infer the latent community structure of nodes from graph structures and persistent homology to encode higher-order information. Authors also theoretically study the entry-wise bound and asymptotic normality of the estimated edge probability matrix to quantify the uncertainty in statistical inference of the edge probabilities. The paper further shows various experiments with the proposed model for the link prediction task.

**Audience:**

Yes

**Broader Impact Concerns:**

N/A There are no concerns regarding the ethical implications of this work.

**Claims And Evidence:**

Yes

**Requested Changes:**

Please see above section for details.

**Strengths And Weaknesses:**

The Introduction section needs some revision as many of the references are much older or even outdated work. Most of the references are 2020 or earlier. Furthermore, several important references are also missing on much more recent and relevant GNN models.

[WWW 2021] Mixed-Curvature Multi-Relational Graph Neural Network for Knowledge Graph Completion. In Proceedings of the Web Conference 2021 (WWW '21). Association for Computing Machinery, New York, NY, USA, 1761–1771. https://doi.org/10.1145/3442381.3450118

[IEEE 2021] "Bi-Level Attention Graph Neural Networks," in 2021 IEEE International Conference on Data Mining (ICDM), Auckland, New Zealand, 2021, pp. 1126-1131, doi: 10.1109/ICDM51629.2021.00133.

The authors summarize their contributions, however it would help to more clearly specify how their contributions are different from existing works e.g., the GCN component has already been proposed previously. It is not so clear that there is novelty in the work.

Furthermore, even in the space of knowledge graphs, works have investigated using non-Euclidean geometric space to model the topological structures of cycles and hierarchy. For example, here is a recent work the authors should reference for relevance:
[KDD 2022] Dual-Geometric Space Embedding Model for Two-View Knowledge Graphs. In Proceedings of the 28th ACM SIGKDD Conference on Knowledge Discovery and Data Mining (KDD '22). Association for Computing Machinery, New York, NY, USA, 676–686. https://doi.org/10.1145/3534678.3539350

I am curious why the authors do not consider modeling the different topologies in non-Euclidean spaces?

Experiment results are comprehensive and show non-trivial gains over the state-of-the-art.

---

> ### Author Response · Authors · 2025-05-14
> **Rebuttal by Authors**
>
> Dear Reviewer LToV, thanks very much for your valuable feedback on our submission. Below is our responses to the concerns you raised. We have also incorporated these contents into the updated version.
>
> **Q1:** The Introduction section needs some revision as many of the references.
>
> **A:** Thank you for raising this concern. We have already added these important and state-of-the-art references in the revision.
>
> **Q2:** Summarize contributions.
>
> **A:** Thanks for your comment and suggestion.
>
> The main contributions of our paper are as follows:
> We propose SBM-TNN, the first model to integrate statistical modeling via Stochastic Block Models (SBMs) with topological learning for graph representation. To enhance the expressiveness of node embeddings, our model incorporates persistent homology and multi-view convolutional layers to extract high-order topological features—an aspect not captured by conventional GCNs. Additionally, we establish theoretical guarantees by deriving entrywise error bounds and proving the asymptotic normality of the estimated edge probability matrix. These results enable the construction of confidence intervals and provide a principled framework for uncertainty quantification, a feature largely absent in existing GNN models.
>
> Compared with [1], we use spectral methods instead of MCMC, and we have the asymptotic distribution of link prediction. Compared with [2], we use spectral methods; however, [2] uses Bayesian methods and only focuses on node classification. Compared with [3], it focuses on node classification, however, our paper focuses on link prediction and knowledge graph completion. For our proposed SBM-TNN model, we use topological neural networks (TNN) instead of GNNs, which allows us to incorporate both local and global topological information of a graph using persistent homology. Equipped with SBM, our model can encode both topological features and community information.
>
> [1] Mehta N, Duke L C, Rai P. Stochastic blockmodels meet graph neural networks; International Conference on Machine Learning. PMLR, 2019: 4466-4474.
>
> [2] Liu X, Yang B, Song W, et al. A block-based generative model for attributed network embedding; World Wide Web, 2021, 24: 1439-1464.
>
> [3] He D, Liang C, Liu H, et al. Block modeling-guided graph convolutional neural networks; Proceedings of the AAAI conference on artificial intelligence. 2022, 36(4): 4022-4029.
>
> **Q3:** Recent work the authors should reference for relevance.
>
> **A:** Thank you for bringing this valuable reference to our attention. We have incorporated the suggested paper into the appropriate sections of the paper.
>
> **Q4:** Different topologies.
>
> **A:** Thank you for this insightful question. We have conducted other topologies including simplicial complexes (i.e., we use both the 1-Hodge Laplacian and graph Laplacian) and metric-induced topologies (i.e., we incorporate features such as shortest path length and Jaccard distance into original node features) for graph link prediction tasks. The experimental results on Cora-ML, Citeseer, and PubMed are:
>
> ||Cora-ML|Citeseer|PubMed|
> |-|-|-|-|
> |SBM-SC| 94.6±0.7 | 96.2±0.2| 95.8±0.4|
> |SBM-MIT| 93.9±0.4 | 93.4±0.3| 94.5±0.6|
> |SBM-TNN| **96.2±0.2** | **97.1±0.3**| **98.2±0.1**|
>
> Compared to SBM-SC (SBM with simplicial complex) and SBM-MIT (SBM with metric-induced topologies), we observe that our SBM-TNN always achieves competitive performances on all datasets.

---

### Decision · Action_Editor_oKFr · 2025-07-02

**Recommendation:** Accept as is

**Audience:**

Yes

**Audience Explanation:**

All reviewers agreed that the work is of interest. The paper sits at the intersection of graph representation learning, topological data analysis and statistical network models — three active research areas for the TMLR community.

The combination of higher-order topology with block-model priors addresses practical demands for interpretable uncertainty in link prediction, and the empirical section demonstrates relevance to both static and dynamic graphs, as well as knowledge-graph completion.

**Claims And Evidence:**

Yes

**Claims Explanation:**

Two of the three expert reviewers judged that the paper’s empirical results and theoretical analysis convincingly substantiate its claims. The dissenting reviewer expressed reservations about over-claiming and missing baselines. After the rebuttal, the authors (i) toned down the overstatement, (ii) added generative‐GNN baselines (VGAE, ARGE/ARVGE) and an edge-feature GAT comparison, and (iii) reported training-time statistics and further ablations, all of which directly address the main points raised

The empirical gains are consistent across 9 datasets—including citation graphs, Amazon, and three knowledge‐graph benchmarks—while the statistical theory provides entry-wise error bounds and asymptotic normality for the estimated edge-probability matrix, enabling uncertainty quantification.

Some clarity issues (notation density, scalability discussion) remain. Still, the core contributions, i.e., integrating SBM-derived community structure with persistent-homology features, backed by new statistical guarantees, now rest on solid experimental and theoretical support. I therefore consider the evidential standard met.